# Massive Editing for Large Language Models Based on Dynamic Weight Generation

**Wentao Wan,**[*] **Qiqing Lao,**[*] **Zhiwei Xie,**[*] **Hefeng Wu, Runnan Lin, Liang Lin, Keze Wang**[†]
Sun Yat-sen University
{wanwentao93, kezewang}@gmail.com

## Abstract

Knowledge Editing (KE) is a field that studies how to modify some knowledge in Large Language Models (LLMs) at a low cost (compared to pre-training). Currently, performing large-scale edits on LLMs while ensuring the Reliability, Generality, and Locality metrics of the edits remain a challenge. This paper proposes a **M**assive **e**diting approach for LLMs based on dynamic weight **G**eneration (**MeG**). Our MeG involves attaching a dynamic weight neuron to specific layers of the LLMs and using a diffusion model to conditionally generate the weights of this neuron based on the input query required for the knowledge. This allows the use of adding a single dynamic weight neuron to achieve the goal of large-scale knowledge editing. Experiments show that our MeG can significantly improve the performance of large-scale KE in terms of Reliablity, Generality, and Locality metrics compared to existing knowledge editing methods, particularly with a high percentage point increase in the absolute value index for the Locality metric, demonstrating the advantages of our proposed method. Code is available at https://github.com/RodeWayne/MeG-for-Knowledge-Editing.

## 1 Introduction

Large Language Models (LLMs), owing to their robust linguistic comprehension capabilities and flexible utilization of their extensive internal knowledge, can proficiently handle diverse text-based tasks (Petroni et al., 2019; Brown et al., 2020; Chowdhery et al., 2023; Hoffmann et al., 2022; Huang et al., 2025b;a). However, the knowledge stored in LLMs may be incorrect or become outdated over time, which can limit their performance in various related tasks, especially those requiring up-to-date knowledge. To update the knowledge, fine-tuning LLMs directly with new knowledge data leads to catastrophic forgetting (Luo et al., 2023; Ramasesh et al., 2021; Gu et al., 2022), causing LLMs to lose much of the knowledge they originally stored; retraining LLMs with all pre-training data with updated new knowledge can prevent catastrophic forgetting, but the computational costs are prohibitively high. To enable low-cost knowledge updates for LLMs, a growing research field known as **K**nowledge **E**diting (**KE**) or **M**odel **E**diting (**ME**) has emerged (Yao et al., 2023b). An ideal KE approach should satisfy three key properties (Wang et al., 2024b; Zhang et al., 2024): a) **Reliability**, all intended new knowledge can be accurately incorporated into LLMs; b) **Generality**, LLMs can correctly respond to equivalent paraphrased queries about the edited knowledge; c) **Locality**, unrelated existing knowledge in LLMs remains unaffected by the edits.

A prevalent knowledge editing scenario involves batch updates - integrating a substantial volume of newly accumulated knowledge (still several orders of magnitude smaller than the original pretraining corpus) into LLMs in a single operation (Gu et al., 2024; Yao et al., 2023b). Existing large-scale editing methods (e.g., MEMIT (Meng et al., 2023), MALMEN (Tan et al., 2024)) exhibit severe performance degradation as the number of edits grows (especially beyond tens of thousands).

Existing large-scale knowledge editing methods are all inner-weight modified methods, which modify some weights inside LLMs to edit knowledge. We analyze that those methods may suffer from several limitations: 1.**Relatively limited knowledge capacity**: By modifying only selected subsets of LLM

---

[*]Equal Contribution
[†]Corresponding Author

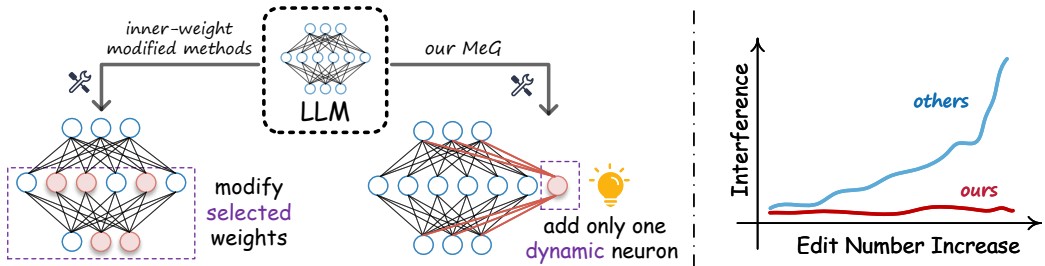

Figure 1: Comparison of our MeG with other LLMs weight modified KE methods.

weights (Gupta et al., 2025), these approaches hit an inherent upper bound on the editable knowledge volume, causing both Reliability and Generality to deteriorate beyond certain scaling thresholds; 2. **Interference Accumulation**: The growing number of weight modifications progressively disrupts the model's original functionality, making it increasingly difficult to maintain Locality as edit scales expand (Huang et al., 2024; Gu et al., 2024). Although neural expansion methods such as T-Patcher (Huang et al., 2023) and SCEN (Yao et al.) - which store new knowledge through additional neurons or expert weights - may potentially overcome the aforementioned limitations, their rapidly growing computational and spatial overhead with increasing edit scales renders them unsuitable for large-scale knowledge editing scenarios.

To alleviate these limitations, inspired by diffusion-based weight generation approaches (Wang et al., 2024a; Soro et al., 2024; Li et al., 2024b), we propose **MeG**, a novel **M**assive knowledge **e**diting method for LLMs based on diffusion-based dynamic weight **G**eneration. Specifically, we train a diffusion model to generate weights of a single neuron conditioned on each knowledge query waiting for editing. Then this neuron is added to the suitable layer of LLMs for the corresponding knowledge query, and the LLMs (now with the additional neuron) execute the inference to produce responses for this query. As illustrated in Fig. 1, our design offers two key advantages: (1) **Higher knowledge capacity** - By utilizing the dynamic neuron mounting based on weight generation, our MeG can edit larger volumes of knowledge, and its knowledge capacity is not affected by the size of the LLMs to be edited; (2) **Scale-invariant interference** - only one neuron is added regardless of edit scale, preventing cumulative interference to LLMs which helps to maintain Locality while large-scale editing.

To further enhance the Generality KE metric, we collect a set of pseudo-knowledge with equivalent expressions and train a text encoder using InfoNCE contrastive loss (Oord et al., 2018). This ensures that queries containing equivalent expressions and their original counterparts are encoded into similar representations, which serve as the condition of our diffusion model. Furthermore, we introduce an entropy-based **Familiarity Network** that performs binary routing between new and existing knowledge, further significantly improving Locality performance. Massive editing experiments on two popular KE datasets, ZsRE (Levy et al., 2017) and COUNTERFACT (Meng et al., 2022), show that our MeG achieves sate-of-art comprehensive KE performance considering Reliability, Generality, and Locality metrics from 1024 to 10k edits, especially with Locality performance significantly outperforming other methods.

## 2 RELATED WORK

**Knowledge Editing** Knowledge Editing (KE) methods fall into four categories: memory storage, localization-editing, meta-learning, and extra-neuron addition. SERAC (Bruel & Sabatier, 2020) uses memory structures for dynamic knowledge retrieval, while ROME (Meng et al., 2022) treats KE as constrained optimization to update FFN layer neurons. MEMIT (Meng et al., 2023) and PMET (Li et al., 2024a) extend this to multi-knowledge edits. Meta-learning methods like MEND (Mitchell et al., 2022) employ hypernetworks for weight updates, with MALMEN (Tan et al., 2024) enabling parallel large-scale edits. Extra-neuron methods include T-Patcher (Huang et al., 2023), which appends trainable neurons to FFN layers, and RASE (Han et al., 2023)/SCEN (Yao et al.), which store weights in external banks. These scale poorly due to limited static storage. Our MeG method replaces static storage with a conditional weight generator, enabling scalable editing with dynamic neuron

weight generation. Since this work focuses on large-scale batch editing scenarios (Yao et al., 2023a), we primarily compare our method in the main text with approaches that target similar scenarios, such as MEMIT, PMET, and MALMEN. For recent representative methods in sequential editing scenarios (Yao et al., 2023a), such as AlphaEdit (Fang et al., 2025) and RLEdit (Li et al., 2025), we provide comparative discussions with these methods in App. A.7.

**Weight Generation**  Neural network weights are typically optimized on data using the Stochastic Gradient Descent (SGD) algorithm (Robbins & Monro, 1951). A particular area of research considers generating weights directly using another neural network (Ha et al., 2017). In the field of meta-learning, a hypernetwork is commonly employed to generate weights for a target network performing new tasks based on the cross-task generalization ability of weights learned from sampled tasks (Schürholt et al., 2022; Chen & Wang, 2022). The Meta-Learing methods generally produce initial weight values, which require further optimization on the current task (Beck et al., 2023; Hospedales et al., 2021). (Wang et al., 2024a) implements the direct generation of final model weights using a diffusion model. Inspired by those works, we introduce the concept of weight generation based on the diffusion model into the task of KE, successfully implementing large-scale knowledge editing and achieving good performance.

## 3 PRELIMINARIES

KE aims to edit a batch of knowledge into LLMs in a low-cost manner, enabling generalization to relevant knowledge without affecting the response of LLMs to irrelevant knowledge. We can denote the targeted editing knowledge set as $(X_e, Y_e)$ with size $N$, and $(x_e, y_e)$ represents one of the $N$ pieces of targeted editing knowledge, where $x_e$ represents the original knowledge query. We use $x_{eq}$ to represent the equivalent statement of $x_e$. All $x_{eq}$ form the set $X_{eq}$. The irrelevant knowledge set can be denoted as $(X_u, Y_u)$, and $(x_u, y_u)$ represents one irrelevant knowledge inside. The original LLM can be denoted as $f_\theta$ with $\theta$ as the weights, and the post-edit LLM can be represented as $f_{\theta_e}$. An ideal knowledge edit can be represented as:

$$f_{\theta_e}(x) = \begin{cases} y_e & \text{if } x \in X_e, \\ y_e & \text{if } x \in X_{eq}, \\ f_\theta(x) & \text{if } x \in X_u. \end{cases} \tag{1}$$

The formal representations corresponding to the 3 main metrics of KE are as follows:

**Reliability** is measured as the average accuracy of the post-edit LLM on original statements of knowledge to be edited using $\mathbb{E}_{x_e \sim X_e} \mathbf{1}\{\arg\max_y f_{\theta_e}(y \mid x_e) = y_e\}$.

**Generality** is measured as the average accuracy of the post-edit LLM on equivalent statements of knowledge to be edited using $\mathbb{E}_{x_{eq} \sim X_{eq}} \mathbf{1}\{\arg\max_y f_{\theta_e}(y \mid x_{eq}) = y_e\}$.

**Locality** is measured as the average accuracy of the post-edit LLM on irrelevant knowledge queries using $\mathbb{E}_{x_u \sim X_u} \mathbf{1}\{f_{\theta_e}(y \mid x_u) = f_\theta(y \mid x_u)\}$.

## 4 METHODOLOGY

As Fig. 2 illustrates, our MeG consists of 4 main components: a **InfoNCE-Tuned Text Encoder**, which is fine-tuned in an InfoNCE contrastive loss (Oord et al., 2018), a **Familiarity Network** routing irrelevant or editing knowledge queries, a **Weight Generation Mechanism** including knowledge-weight collection and a Weight-Generation model based on Diffusion Transformer (DiT) (Peebles & Xie, 2023), and a **Single Dynamic-weight Neuron Attaching** mechanism for large-scale KE.

### 4.1 COMPONENT DETAILS

**a. InfoNCE-Tuned Text Encoder**  Knowledge queries require representation obtained through a Text Encoder $f_{TE}$ before entering Familiarity Networks or the weight-generation model. We use a pre-trained BERT (Devlin et al., 2019) as the architecture and initial weights of the Text Encoder, employing the CLS token embedding from the output end as the aforementioned representation. Here, it is necessary to further fine-tune the Text Encoder to narrow the gap between the representations

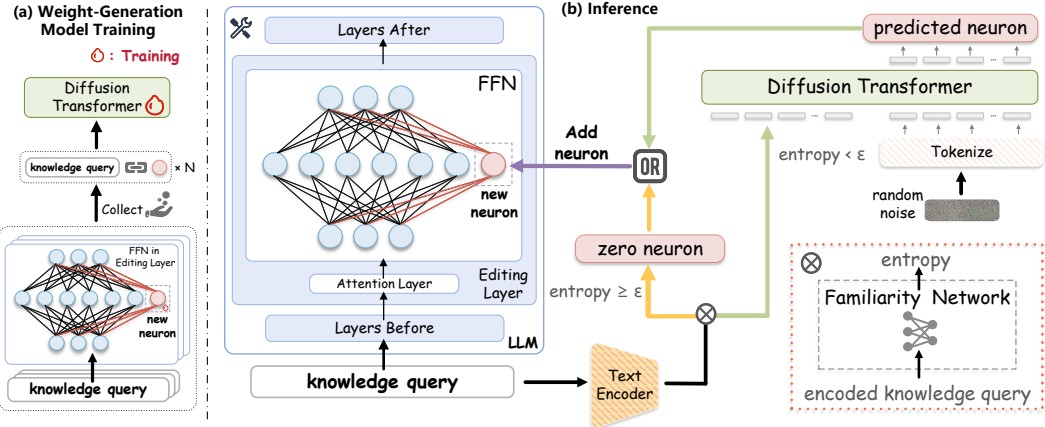

Figure 2: Overall pipeline of our MeG.

of the original and equivalent expressions of the same target editorial knowledge. This adjustment facilitates similar behaviors of the equivalent and original expressions in the Familiarity Networks or the weight-generation model, thereby enhancing performance on the Generality metric.

We consider the aforementioned issue as a contrastive representation learning problem. For any equivalent expression $x_{eq}^i$, its corresponding original expression $x_e^i$ is considered a positive sample. The original expressions of other knowledge $x_e^j$ to be edited are considered negative samples. We employ the InfoNCE loss below to fine-tune $f_{TE}$:

$$L_{f_{TE}} = -\frac{1}{B}\sum_{i=1}^{B}\log\frac{\exp\left(\mathrm{sim}(x_{eq}^i, x_e^i)/\tau\right)}{\sum_{j=1}^{B}\exp\left(\mathrm{sim}(x_{eq}^i, x_e^j)/\tau\right)}, \tag{5}$$

where $B$ is the batch size while training, $sim()$ calculates the cosine similarity, and $\tau$ is a temperature hyperparameter used to control the smoothness of the distribution. Since we are unable to obtain real equivalent expressions, we have additionally gathered a batch of original and equivalent expressions of pseudo-edited knowledge to train the Text Encoder. Our experiments demonstrate that this data strategy exhibits good generalization capabilities.

**b. Familiarity Network** The motivation for the Familiarity Network is to rely only on accessible edited knowledge to train and determine whether the current query is irrelevant knowledge. To solve this problem, we creatively used the following mechanism: the training of a neural network is an entropy reduction process, so when the network uses data or neighboring data seen during training as input, the entropy of its output distribution is much lower than that of unseen data. Specifically, the Familiarity Network $f_\mu$ is a k-classifier ($K << N$, we adopt $K = 10$) network with 5 FFN blocks of fully connected layer followed by a final classification head layer. First we collect all the targeted editing knowledge queries $X_e$, and assign an random category label $y_c$ from 0 to $K-1$ to each $x_e$, construct the train set $\{x_e^i, y_c^i\}_{i=1}^N$ of $f_\mu$. Training $f_\mu$ on the above train set using the classifier Cross-Entropy loss:

$$L_{f_\mu} = -\sum_i^N \log[f_\mu(f_{TE}(x_e^i))]_{index(y_c^i)}, \tag{6}$$

where $f_{TE}$ is the above-mentioned Text Encoder which has been fine-tuned and is frozen while training $f_\mu$. After training $f_\mu$ until convergence, we calculate the entropy at the Familiarity Network output for any knowledge query. It is found that the entropy for queries involving irrelevant knowledge is significantly higher than that for queries related to the original or equivalent expressions of the edited knowledge.

**c. Weight Generation Mechanism** This mechanism includes Knowledge-Weight Collection and a Weight-Generation model. Since our weight generation network utilizes a diffusion model, we do not train the weight generation network end-to-end in a manner similar to meta-learning. Instead, we first

collect a dataset of the knowledge items to be edited and their corresponding added neuron-weight pairs, then train the weight generation model on this dataset.

We find that the impact of fine-tuning by adding a single neuron to FFN in different layers of LLMs varies significantly, and we diverge from the T-Patcher (Huang et al., 2023) method, which adds neurons in the last FFN. Instead, for different LLMs, we first explore and select specific FFN layers (please refer to Sec. 5.3 for details) in which to add that dynamic weight neuron.

**c.1 Knowledge-Weight Collection**    As illustrated in the middle part of Fig. 2, when we have selected to add a new neuron with weight $w_e$ at FFN in $Block_i$. For each original expression $x_e$ of an editing knowledge, input $x_e$ into the LLM $f_{\theta_e}$, we get $y = f_{\theta_e}(x_e) = f(x_e; (\theta, w_e))$. Make $\theta$ be frozen, optimizing $w_e$ to make $y = y_e$, then we get one knowledge-weight pair $(x_e, w_e)$. Perform the above operations on the original expressions of all N knowledge items to be edited, thereby collecting N pairs of knowledge and weights $(x_e^i, w_e^i)_{i=1}^N$.

**c.2 Weight Generation Model**    We use the DiT architecture diffusion model to conduct the text-to-weight generation. Cause the approximation of the dimensions of $w_e$ and an image, we refer to the standard text-to-image generation procedure of DiT for the weight of the added neuron generation. At the input stage, we use the knowledge-weight pairs $(x_i^e, w_i^e)$ collected above. Each $w_i^e$ is split into non-overlapping patches of size 100 to form the input tokens.

For conditioning, we use the InfoNCE-Tuned Text Encoder. The editing knowledge query $x^e$ is passed through the encoder, and the [CLS] token embedding is used as the condition input to the diffusion model.

To improve training stability and performance, we adopt the **velocity prediction** (v-prediction) (Salimans & Ho, 2022) formulation of the diffusion objective. Instead of directly predicting the noise or the original data, the model learns to predict the velocity $\mathbf{v}_t$, defined as a linear combination of the clean target $w$ and the added noise $\boldsymbol{\epsilon}$:

$$\mathbf{v}_t = \alpha_t \cdot \boldsymbol{\epsilon} - \beta_t \cdot w \tag{7}$$

Here, $\alpha_t$ and $\beta_t$ are coefficients determined by the noise schedule. The model is trained to minimize the following MSE loss:

$$\mathcal{L}_{\text{v-pred}} = \mathbb{E}_{w,c,t} \left[ \|\mathbf{v}_t - \hat{\mathbf{v}}_\theta(w_t, t, c)\|_2^2 \right] \tag{8}$$

where $w_t$ is the noisy version of the target weight at timestep $t$, $c$ is the condition (CLS embedding), and $\hat{\mathbf{v}}_\theta$ is the model prediction of the velocity.

## 4.2    INFERENCE PROCEDURE

**Single Dynamic-weight Neuron Attaching**    As illustrated in the right part of Fig. 2, in the stage of Inference, first, the current knowledge query $x$ is input into the Text Encoder $f_{TE}$ to get the representation $z = f_{TE}(x)$. Then we put $z$ into the Familiarity Network to get the output distribution $P_\mu = f_\mu(z)$. Calculating the entropy through the above distribution as

$$H = -\sum_{k=1}^{K} P_\mu^k \log P_\mu^k, \tag{9}$$

where $K$ is the output dimension of $f_\mu$. We compare $H$ with a hyperparameter $\epsilon$, if $H < \epsilon$, means the original input $x$ is the original or equivalent expression of the knowledge to be edited, else it is irrelevant knowledge. Then the corresponding generated weight is produced as:

$$w_e = \begin{cases} \mathbf{0} & \text{if } H \geq \epsilon, \\ f_D(z) & \text{if } H < \epsilon, \end{cases} \tag{10}$$

where $f_D(z)$ means the DiT trained before, and with current knowledge query representation $z$ as the condition. We further utilize the generation capability of the DiT model to produce the corresponding weight $w_e$. Specifically, DiT performs the reverse denoising process to iteratively generate $w_e$ conditioned on the textual representation $z$. The process can be expressed as:

$$w_e = \frac{1}{\sqrt{\bar{\alpha}_t}} \left( w_t - \sqrt{1 - \bar{\alpha}_t} \cdot v \right), \tag{11}$$

where $w_t$ is the intermediate noisy weight at timestep $t$, $\bar{\alpha}_t$ is the cumulative product of noise schedule coefficients, and $v$ is the velocity predicted by the model. The final predicted weight $w_e$ is obtained after iterating this reverse process from $t = T$ to $t = 0$.

In practice, we compress the generation steps from the original 1000 to only 50 steps using fast sampling techniques. Our experiments show that this acceleration maintains comparable performance in terms of weight generation quality, while significantly improving inference efficiency (refer to 6.3).

After the weight of the added neuron has been produced, we load the weight into this neuron, then get the final response to current knowledge query $x$ of post-edit LLM through

$$y = f_{\theta_e}(x) = f(x; \{\theta, w_e\}). \tag{12}$$

## 5 EXPERIMENTS

### 5.1 EXPERIMENTS SETUP

**Datasets and LLMs** We conduct edits on two datasets: the Zero-Shot Relation Extraction (ZsRE) dataset (Levy et al., 2017) for error correction and the COUNTERFACT dataset (Meng et al., 2023) for counterfactual updates. To ensure validity, we apply rigorous preprocessing to both datasets. Detailed preprocessing procedures are provided in App. A.9. All baseline methods are evaluated under the same preprocessed datasets. Experiments are conducted on three large language models: Phi-2 (2.7B) (Javaheripi et al., 2023), GPT-J (6B) (Wang & Komatsuzaki, 2021), and Llama-3 (8B) (Dubey et al., 2024) in Float16.

**Evaluation Metric** Following (Yao et al.), we adopt 3 evaluation metrics: Reliability, Generality, and Locality. For each metric, we report results under two generation settings: besides the Teacher Forcing generation (TF) setting used by most existing approaches, we also adopt a more realistic Prefix-autoregressive generation (AG) setting. In previous works, the Locality metric considers the change in accuracy on a specific dataset before and after editing, where the accuracy is extremely low (below 30% ). This does not assess whether the response to the knowledge the LLM originally got wrong on that dataset has changed or not. We believe that a metric assessing whether the response of LLMs to irrelevant knowledge remains the same after KE better reflects the original consideration of the Locality metric. Hence, we use the Equation of Locality in Sec. 3 to calculate the Locality metric. To evaluate the comprehensive performance, following (Meng et al., 2023), we use the Score metric, which is defined as the harmonic mean of the 6 results of the afore-mentioned 3 evaluation metrics (Reliability, Generality, and Locality under TF and AG generation settings).

**Baseline** The methods we have chosen for comparison include: direct fine-tuning (FT) in which we directly fine-tune the FFN components on the selected layer of our MeG in the LLM with all the edited knowledge; two locate-then-edit methods for massive KE, MEMIT (Meng et al., 2023), and PMET (Li et al., 2024a); a massive KE method based on Meta-Learning, MALMEN (Tan et al., 2024); SCEN (Yao et al.), a KE method based on adding or routing new neuron weights. For all those baseline methods, we implement them on Phi-2 (Javaheripi et al., 2023), GPT-J (Wang & Komatsuzaki, 2021), and Llama-3 (Dubey et al., 2024) based on their official codes. Details can be found in App. A.12.

### 5.2 EXPERIMENTAL RESULTS AND ANALYSES

**Scaling Curves** Fig. 3 shows the performance curves of the baseline methods and our MeG for Phi-2 and GPT-J on ZsRE and COUNTERFACT datasets while especially editing 1024, 2048, 4096, and 10000 knowledge items. (Due to space constraints, we have placed the overall performance curve of the Llama-3 model in App. A.1.) Here we exhibit the Score metric of the results (please refer to "Evaluation Metric" in Sec. 5.1). Due to prohibitive computational and memory overheads in SCEN that scale rapidly with edit size, we evaluate SCEN only up to 2048 edits. For more detailed results on each metric, please refer to App. A.2. As shown in the figure, our MeG consistently outperforms all baseline methods across all models and datasets under four different edit scales. Notably, as the edit scale increases from 1024 to 10000, MeG exhibits a slower performance degradation, further widening its performance gap over competing methods. This robust scalability strongly validates the superiority of our MeG in large-scale editing tasks.

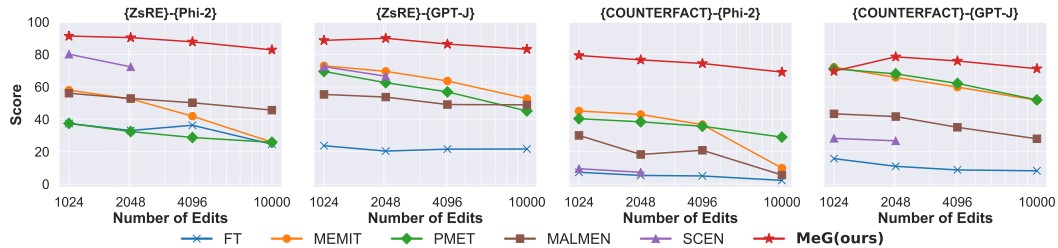

Figure 3: Performance for Phi-2 (2.7B) and GPT-J (6B) on two datasets while scaling the editing. Here, Score is defined as the harmonic mean of 6 results of 3 evaluation metrics - Reliability, Generality, and Locality under TF and AG generation settings.

Table 1: Performance comparison on Phi-2 (2.7B), GPT-J (6B), and Llama-3 (8B) models for ZsRE (edit_num=10000) across Reliability, Generality, Locality, and overall Score.

| Model | Method | Reliability ↑ | | Generality ↑ | | Locality ↑ | | Score ↑ |
|---|---|---|---|---|---|---|---|---|
| | | AG | TF | AG | TF | AG | TF | |
| **Phi-2** | FT | 69.27 | 80.09 | 45.96 | 67.10 | 6.27 | 48.88 | 24.64 |
| | MEMIT | 65.88 | 81.18 | 35.69 | 59.84 | 7.11 | 53.40 | 25.91 |
| | PMET | 20.09 | 47.82 | 15.02 | 43.13 | 17.73 | 64.62 | 25.83 |
| | MALMEN | 71.79 | 85.94 | 41.54 | 68.67 | 18.22 | 80.78 | 45.64 |
| | **MeG (ours)** | **95.07** | **97.04** | **59.69** | **74.17** | **91.14** | **95.84** | **82.80** |
| **GPT-J** | FT | **100.0** | **99.80** | **100.0** | **99.80** | 4.72 | 40.27 | 21.68 |
| | MEMIT | 88.11 | 94.22 | 62.50 | 78.59 | 20.30 | 71.93 | 52.70 |
| | PMET | 74.47 | 86.87 | 55.02 | 72.42 | 16.31 | 67.96 | 45.13 |
| | MALMEN | 96.78 | 98.54 | 59.35 | 79.09 | 16.56 | 81.41 | 48.92 |
| | **MeG (ours)** | 99.11 | 99.16 | 61.69 | 75.68 | **83.99** | **94.20** | **83.20** |
| **Llama-3** | FT | **100.0** | **99.87** | 54.41 | **90.06** | 13.48 | 54.01 | 42.20 |
| | MEMIT | 75.68 | 88.30 | 50.88 | 71.87 | 10.51 | 54.89 | 34.99 |
| | PMET | 65.40 | 80.40 | 54.58 | 72.86 | 20.65 | 64.22 | 48.48 |
| | MALMEN | 87.23 | 94.72 | 57.95 | 80.92 | 44.65 | 85.86 | 70.03 |
| | **MeG (ours)** | 98.90 | 99.44 | **61.33** | 78.95 | **85.02** | **94.23** | **83.90** |

Table 2: Performance comparison on Phi-2 (2.7B), GPT-J (6B), and Llama-3 (8B) models for COUNTERFACT (edit_num=10000) across Reliability, Generality, Locality, and overall Score.

| Model | Method | Reliability ↑ | | Generality ↑ | | Locality ↑ | | Score ↑ |
|---|---|---|---|---|---|---|---|---|
| | | AG | TF | AG | TF | AG | TF | |
| **Phi-2** | FT | 53.49 | 53.49 | 14.15 | 14.14 | 0.43 | 11.98 | 2.32 |
| | MEMIT | 78.38 | 78.29 | 33.61 | 33.35 | 2.03 | 37.50 | 9.92 |
| | PMET | 50.29 | 49.95 | 21.02 | 20.84 | 18.13 | 61.48 | 29.00 |
| | MALMEN | 92.48 | 92.43 | 19.26 | 19.05 | 1.22 | 8.60 | 5.65 |
| | **MeG (ours)** | **97.67** | **97.78** | **48.50** | **51.24** | **72.36** | **80.51** | **69.09** |
| **GPT-J** | FT | **99.99** | **99.99** | 59.55 | 59.55 | 1.83 | 7.84 | 8.25 |
| | MEMIT | 95.56 | 95.55 | 54.62 | 54.42 | 26.22 | 49.35 | 51.72 |
| | PMET | 96.65 | 96.62 | **66.82** | **66.74** | 23.47 | 44.92 | 51.94 |
| | MALMEN | 99.97 | 99.97 | 26.44 | 26.44 | 13.80 | 21.56 | 27.97 |
| | **MeG (ours)** | 99.00 | 99.01 | 61.04 | 61.84 | **61.50** | **65.52** | **71.19** |
| **Llama-3** | FT | **99.99** | **99.99** | 24.67 | 25.30 | 0.68 | 4.92 | 3.38 |
| | MEMIT | 86.48 | 86.23 | 39.28 | 39.68 | 9.57 | 33.00 | 28.76 |
| | PMET | 89.25 | 89.12 | **54.26** | **54.08** | 14.15 | 44.16 | 39.30 |
| | MALMEN | 92.26 | 92.17 | 28.75 | 29.44 | 3.39 | 23.52 | 14.02 |
| | **MeG (ours)** | 98.48 | 98.50 | 46.06 | 48.17 | **61.01** | **71.79** | **64.45** |

Table 3: Performance comparison of Phi-2 on general benchmarks after editing 10000 knowledge items on ZsRE (the higher result means the better performance). Our MeG achieves competitive performance across all benchmarks.

| Task | Before Edit | FT | MEMIT | PMET | MALMEN | MeG (ours) |
|------|-------------|------|-------|------|--------|------------|
| BBH | 40.58 | 20.12 | 30.31 | 39.88 | 38.45 | **40.67** |
| GSM8K | 42.84 | 32.68 | 19.26 | 60.80 | 48.60 | **61.18** |
| MMLU | 56.98 | 52.68 | 45.19 | 55.53 | 53.97 | **57.00** |
| RTE | 60.79 | 1.53 | 55.44 | 59.62 | 56.89 | **60.47** |
| SST2 | 89.56 | 89.33 | 53.56 | 86.93 | 88.53 | **89.68** |

**Results on 10k Editing** For a fine-grained comparison of model editing methods at scale, Tables 1 and 2 present the performance on Reliability, Generality, and Locality metrics of Phi-2, GPT-J, and Llama-3 models after 10000 edits on the ZsRE and COUNTERFACT datasets, respectively. We utilize the evaluation metrics described in "Evaluation Metric" in Sec. 5.1. As evidenced by the tables, our method demonstrates significantly superior comprehensive performance (Score) across all experimental configurations - spanning both datasets (ZsRE and COUNTERFACT) and model architectures (Phi-2, GPT-J, and Llama-3) - outperforming all baseline methods by considerable margins. Our comparative analysis reveals distinct performance patterns: (1) Baseline methods show significantly lower Reliability and Generality on Phi-2 than GPT-J and Llama-3, which we attribute to Phi-2's smaller model capacity limiting their weight-modification effectiveness; (2) All baselines exhibit notably poor Locality, particularly under AG generation settings. Our method surpasses all baselines on every metric for Phi-2. While showing marginally lower Reliability/Generality than some baselines on GPT-J, it achieves substantial Locality advantages: +63.69% (AG) and +12.79% (TF) over the second-best on ZsRE, and +35.28% (AG) and +16.17% (TF) on COUNTERFACT. The situation on Llama-3 is similar to that on GPT-J.

**Results on General Bench after KE** To evaluate the general ability degradation after large-scale KE, after 10000 edits, we test the post-edited LLMs of different KE methods on several general Benchmarks: BBH (Suzgun et al., 2023) (Complicated Reasoning), GSM8K (Cobbe et al., 2021) (Math Ability), MMLU (Hendrycks et al.) (Cross-disciplinary Capabilities), RTE (Dagan et al., 2005) (Recognizing Textual Entailment), SST2 (Socher et al., 2013) (Sentiment Analysis and Fine-grained Semantic Classification). Table 3 shows the results of Phi-2 after 10000 edits on ZsRE. Table 3 demonstrates that our MeG can maintain superior general capability after large-scale edits compared to baseline approaches. We attribute this advantage to our MeG's effective control over interference during large-scale editing.

## 5.3 Ablation Study

**Effectiveness of Contrastive Representation Learning** To validate the effectiveness of the contrastive learning framework to enhance Generality based on InfoNCE loss for the Text Encoder (TE), which utilizes BERT in our MeG. We compare three TE variants: 1. Frozen BERT: directly uses the pretrained BERT without fine-tuning; 2. MSE-Tuned BERT: fine-tuned with MSE loss on paraphrased pairs; 3. InfoNCE-Tuned BERT (Ours): our contrastive learning approach. We test the Generality performance for GPT-J on the COUNTERFACT dataset, results are recorded in Table 4. As shown in Table 4, MeG under our contrastive learning TE achieves superior Generality than comparative settings, which validates the effectiveness of the contrastive learning framework for TE in our MeG.

Table 4: Generality performance of various TE for {COUNTERFACT}-{GPT-J}-1024 edits.

| TE | | BERT | MSE-Tuned BERT | InfoNCE-Tuned BERT |
|------|------|------|----------------|--------------------|
| Generality | AG | 18.46 | 56.54 | **84.96** |
| | TF | 18.46 | 56.93 | **86.33** |

**Effectiveness of Familiarity Network** We conducted an ablation study of 1024 ZsRE edits on Phi-2 to evaluate the contribution of the Familiarity Network (FN), and the results are presented in Table 5. The table compares the performance across two configurations: without Familiarity Network (w/o FN), and our proposed

Table 5: Performance comparison of {ZsRE}-{Phi-2}-1024 editing. "w/o FN" means "without Familiarity Network".

| Layer | Reliability ↑ | | Generality ↑ | | Locality ↑ | | Score ↑ |
|-------|------|------|------|------|------|------|------|
| | AG | TF | AG | TF | AG | TF | |
| w/o FN | **99.71** | **99.55** | **83.50** | **89.48** | 53.81 | 83.54 | 81.32 |
| Ours | 99.61 | 99.49 | 80.37 | 87.24 | **88.67** | **95.75** | **91.30** |

method (Ours). The results show that without FN, our MeG can achieve a good performance of Locality. And the Familiarity Network contributes substantially to Locality improvement, achieving 34.86% (AG) and 12.21% (TF) performance gains respectively. These results provide conclusive evidence of its effectiveness.

# 6 DISCUSSION

## 6.1 INTERFERENCE ANALYSIS OF ADDING SINGLE NEURON

In Table 6, we present the performance of our MeG without the familiarity network as the edit scale increases, aiming to analyze the interference caused by adding a single dynamically generated neuron to the original LLMs based on the Locality performance. The results reveal: 1. Even without Familiarity Network, MeG surpasses weight-modification-based methods in Locality by a clear margin; 2. Performance on Locality does not exhibit monotonic degradation trends (initial rise followed by mild decline), with only 4.89% (AG) and 3.06% (TF) degradation at 10k vs. 1024 edits — strongly validating our design.

Table 6: Performance of adding a single neuron under different KE sizes for {ZsRE}-{Phi-2}.

| Edit_num | Reliability ↑ | | Generality ↑ | | Locality ↑ | | Score ↑ |
|---|---|---|---|---|---|---|---|
| | AG | TF | AG | TF | AG | TF | |
| 1024 | 99.71 | 99.55 | **83.50** | **89.48** | 53.81 | 83.54 | 81.32 |
| 2048 | **99.80** | **99.63** | 83.01 | 88.91 | 56.20 | 84.25 | **82.18** |
| 4096 | 99.58 | 99.58 | 74.76 | 83.93 | **58.28** | **86.02** | 80.91 |
| 10000 | 95.76 | 97.38 | 64.29 | 77.20 | 48.92 | 80.48 | 73.09 |

## 6.2 WHY USING DIFFUSION MODEL TO GENERATE WEIGHTS

Our MeG utilizes a diffusion model to generate the weights of the added neuron, and the primary motivation for it stems from two considerations.

First, our experiments reveal that the editing performance metrics, particularly Reliability and Generality, are highly sensitive to the precision of the generated neuron weights. Diffusion models have demonstrated strong capabilities in fine-grained generation in text-to-image tasks, making them a natural choice for achieving higher precision in this context. Additionally, in our task, the dimensionality of the neuron weights generated conditionally on text (e.g., 25602 dimensions for Phi-2, 40962 dimensions for GPT-J and Llama-3) is comparable to the dimensionality of a single image. This suggests that diffusion models, which have shown exceptional performance in image generation, hold significant promise for effectively addressing the challenges posed by our task.

Second, our choice is inspired by the emerging trend of "Neural Network Diffusion" in the research community. Recent pioneering works (Wang et al., 2024a; Erkoç et al., 2023), such as p-diff (Wang et al., 2024a), have demonstrated that diffusion models are highly effective at modeling the complex distribution of neural network parameters, outperforming VAE-based hypernetworks. However, unlike previous works that primarily focus on generating entire static model parameters from scratch, we creatively apply the idea of weight generation to a knowledge-query-conditioned single-neuron dynamic weight generation and loading mechanism, thereby overcoming the challenges of increasing additional memory overhead and interference with the original LLM as the scale of knowledge editing expands.

To verify the necessity of our diffusion module, we conduct another ablation experiment, replacing the DiT module with an MLP module while keeping other components unchanged. Table 7 shows the comparison of editing performance between using MLP and DiT as the weight generation network in the Llama-3-ZsRE experimental group, as the KE scale increases from 1024 to 10k.

From the results in Table 7, we can find that while the MLP variant achieves comparable Reliability and Locality, it suffers a significant drop in Generality compared to the DiT-based MeG. This is consistent with our analysis. We also have observed that the MLP variant reduces training time by 90%. This suggests that our framework is flexible: one could opt for an MLP weight generator for extreme efficiency, but DiT is essential for achieving a good Generality performance, which is one of the primary goals of our MeG.

Table 7: Performance comparison of DiT vs MLPs as the weights generation module in our MeG on Llama-3 (8B) models for ZsRE (under various editing scales) across Reliability, Generality, Locality, and overall Score.

| Edit_Scale | Generation_Module | Reliability ↑ | | Generality ↑ | | Locality ↑ | | Score ↑ |
|---|---|---|---|---|---|---|---|---|
| | | AG | TF | AG | TF | AG | TF | |
| 1024 | MLP | 97.66 | 99.09 | 59.57 | 78.42 | **81.64** | **93.18** | 82.36 |
| | DiT | **99.90** | **99.91** | **81.35** | **89.14** | 78.71 | 92.54 | **89.50** |
| 2048 | MLP | 99.27 | 99.67 | 62.30 | 79.46 | **86.13** | **94.83** | 84.63 |
| | DiT | **99.90** | **99.83** | **76.81** | **86.82** | 84.03 | 94.48 | **89.49** |
| 4096 | MLP | 99.83 | 99.79 | 62.67 | 79.55 | **83.52** | **93.82** | 84.27 |
| | DiT | **99.88** | **99.83** | **73.58** | **85.15** | 81.62 | 93.21 | **87.79** |
| 10000 | MLP | 95.84 | 98.64 | 45.49 | 70.82 | **86.73** | **95.04** | 76.21 |
| | DiT | **98.90** | **99.44** | **61.33** | **78.95** | 85.02 | 94.23 | **83.90** |

## 6.3 EFFICIENCY ANALYSIS

Although our MeG achieves outstanding comprehensive performance in large-scale KE, its reliance on a diffusion-based weight generation model raises concerns regarding computational efficiency. This subsection analyzes the time efficiency of our MeG at the inference stage. During the inference phase of our MeG method, the time cost of the diffusion model constitutes the extra overhead. We use DDIM (Song et al., 2021) to accelerate diffusion inference. DDIM is a standard, theoretically rigorous method that generalizes the Markovian diffusion process to a non-Markovian one, allowing for deterministic sampling with significantly fewer steps while mapping the same latent noise to the data distribution. We evaluated the knowledge editing (KE) performance on the ZsRE dataset (scale: 10,000 edits) for the Phi-2 model, along with the time costs of both the diffusion model (under varying denoising steps) and the proportion of total generation cost (including the diffusion generation cost and the LLM generation cost) on a single NVIDIA RTX 4090D GPU. The results are summarized in Table 8. For further efficiency analysis, please refer to App. A.8.

Our analysis shows that for 10,000 knowledge edits, the denoising steps of the diffusion model can be reduced to 50 while largely preserving all KE performance metrics. In this configuration, the average additional time per edit is 0.03s, and the overhead introduced by the diffusion model accounts for only 5.58% of the total time (see Table 8). The choice of 50 inference steps is consistent with the widely shared experience in the diffusion model community (for example, Stable Diffusion (Rombach et al., 2022) also uses 50 steps). In all other experiments of our MeG across different groups, the weight generation diffusion inference process also adopts 50 inference steps, which has consistently maintained strong experimental performance. This demonstrates that MeG imposes a negligible impact on inference-time efficiency, thereby not compromising the practical utility of the method.

Table 8: Performance and inference time cost of our MeG under different diffusion generation steps for {ZsRE}-{Phi-2}-10000 editing.

| Diff_Step | Reliability ↑ | | Generality ↑ | | Locality ↑ | | Score ↑ | Diff_time_each (s) | Proportion |
|---|---|---|---|---|---|---|---|---|---|
| | AG | TF | AG | TF | AG | TF | | | |
| 1000 | 95.07 | 97.04 | **59.69** | **74.17** | 91.14 | 95.84 | **82.80** | 0.304 | 0.3212 |
| 100 | **95.09** | **97.12** | 58.71 | 73.80 | **91.59** | **95.93** | 82.49 | 0.046 | 0.0807 |
| 50 | 94.59 | 96.71 | 59.05 | 73.97 | 91.55 | 95.90 | 82.51 | 0.031 | 0.0558 |
| 10 | 87.84 | 93.07 | 51.97 | 69.88 | 91.08 | 95.84 | 77.83 | 0.020 | 0.0341 |

## 7 CONCLUSION

We propose a large-scale KE method for an LLM based on weight generation for the dynamic weight neuron added to the LLM. Additionally, we employ familiarity networks and contrastive learning techniques to enhance the locality and generality metrics. Experiments on popular KE datasets demonstrate that our method achieves state-of-the-art performance in large-scale KE scenarios.

## 8 ACKNOWLEDGMENTS

This work was supported in part by the National Natural Science Foundation of China (NSFC) under Grant 62276283 and 62272494, in part by the China Meteorological Administration's Science and Technology Project under Grant CMAJBGS202517, in part by Guangdong-Hong Kong-Macao Greater Bay Area Meteorological Technology Collaborative Research Project under Grant GHMA2024Z04, in part by Fundamental Research Funds for the Central Universities, Sun Yat-sen University under Grant 23hytd006, in part by Guangdong Provincial High-Level Young Talent Program under Grant RL2024-151-2-11, in part by the Key Development Project of the Artificial Intelligence Institute, Sun Yat-sen University, and in part by The Major Key Project of PCL (Grant No. PCL2025A17).

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

## 9 ETHICS STATEMENT

This work adheres to the ICLR Code of Ethics. Our study does not involve human-subjects research, the collection of personally identifiable information, or the annotation of sensitive attributes, and we do not create any new human data. All experiments are conducted exclusively on publicly available, widely used Knowledge Editing benchmarks (e.g., ZsRE, COUNTERFACT), strictly under their respective licenses and terms of use.

## 10 REPRODUCIBILITY STATEMENT

We take reproducibility seriously. To facilitate replication, we describe each component design of our method and evaluation metrics in our experiments in detail. We report the detailed data preprocessing protocol(App. A.9) and implementation details (App. A.11), including our method and the baseline methods(prompts, hyperparameters, etc.). We also provide ablations isolating the contribution of each component (Sec. 5.3). All experiments are averaged over five independent runs with fixed random seeds to ensure robustness, and results are reported on standard benchmarks. Together, these practices ensure that our findings can be reliably reproduced by the community.

## A APPENDIX

### A.1 PERFORMANCE CURVES FOR LLAMA-3

Due to space limitations, we present the performance curve of the Llama-3 model here, as shown in Fig. 4. From the results, it can be seen that the overall performance of the Llama-3 model exhibits the same trend as on the Phi-2 and GPT-J models, with overall performance surpassing other methods. As the scale of editing increases, the performance gap between it and other methods becomes even larger.

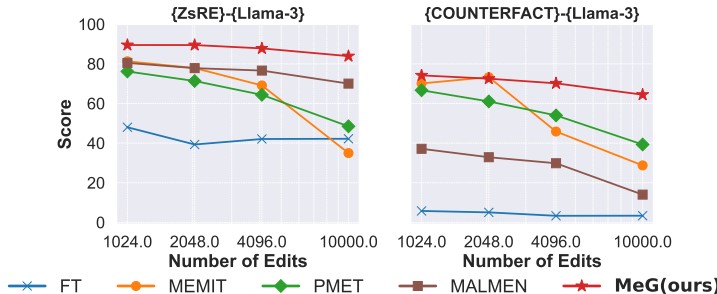

Figure 4: Performance for Llama-3 (8B) on two datasets while scaling the editing. Here, Score is defined as the harmonic mean of 6 results of 3 evaluation metrics - Reliability, Generality, and Locality under TF and AG generation settings.

### A.2 MORE RESULTS ON DIFFERENT EDITING SCALE (1024, 2048, 4096)

Fig. 3 and Fig. 4 illustrate the comprehensive editing performance of different methods on Phi-2, GPT-J, and Llama-3 models over the ZsRE and COUNTERFACT datasets, with Score reflecting the trend as edit scaling. Score is defined as the harmonic mean of 6 results of 3 evaluation metrics - Reliability, Generality, and Locality. Here we exhibit the detailed performance results of each metric from 1024 to 4096 in Table 9 to 14. For results of 10000 edits, please refer to Table 1 and 2 in our main script. The experimental results in the above table demonstrate that our method achieves the best overall performance across all edit scales. While we occasionally underperform the top method on specific metrics (e.g., Reliability or Generality), our approach basically dominates others in Locality by a significant margin. This superior scalability stems from our method's unique design of introducing only a single dynamic neuron to the LLM, which effectively confines interference without causing cumulative effects as editing scale increases.

Table 9: Performance comparison on Phi-2, GPT-J, and Llama-3 models for ZsRE dataset (edit_num=1024) across Reliability, Generality, Locality, and overall score.

| Model | Method | Reliability ↑ | | Generality ↑ | | Locality ↑ | | Score ↑ |
|---|---|---|---|---|---|---|---|---|
| | | AG | TF | AG | TF | AG | TF | |
| Phi-2 | FT | 99.51 | 99.61 | 83.49 | 91.04 | 10.15 | 53.13 | 37.41 |
| | MEMIT | 96.97 | 98.25 | 69.34 | 84.10 | 23.63 | 69.38 | 57.95 |
| | PMET | 35.64 | 60.42 | 21.58 | 50.18 | 28.12 | 72.77 | 37.46 |
| | MALMEN | **100.0** | **100.0** | **84.27** | **93.01** | 19.14 | 82.25 | 56.06 |
| | SCEN | 96.97 | 98.08 | 54.49 | 66.64 | **96.77** | 93.61 | 80.13 |
| | **MeG (ours)** | 99.61 | 99.49 | 80.37 | 87.24 | 88.67 | **95.75** | **91.30** |
| GPT-J | FT | **100.0** | 99.83 | **100.0** | **99.83** | 5.27 | 43.20 | 23.72 |
| | MEMIT | 96.88 | 98.51 | 77.44 | 87.35 | 39.26 | 83.80 | 72.96 |
| | PMET | 87.70 | 94.24 | 73.14 | 85.33 | 37.40 | 82.13 | 69.51 |
| | MALMEN | **100.0** | **100.0** | 85.64 | 93.29 | 18.45 | 84.47 | 55.33 |
| | SCEN | 94.62 | 96.19 | 52.63 | 65.15 | 62.79 | 86.33 | 72.44 |
| | **MeG (ours)** | 99.41 | 99.25 | 83.59 | 89.09 | **75.29** | **90.35** | **88.66** |
| Llama-3 | FT | **100.00** | **99.91** | 45.21 | 87.84 | 18.26 | 59.83 | 48.00 |
| | MEMIT | 94.53 | 97.48 | 80.96 | 88.76 | 55.57 | 88.00 | 81.29 |
| | PMET | 83.79 | 91.53 | 75.49 | 85.76 | 51.76 | 85.51 | 76.16 |
| | MALMEN | 99.31 | 99.77 | **85.05** | **94.16** | 48.04 | 88.68 | 80.47 |
| | **MeG (ours)** | 99.90 | **99.91** | 81.35 | 89.14 | **78.71** | **92.54** | **89.50** |

Table 10: Performance comparison on Phi-2, GPT-J, and Llama-3 models for ZsRE dataset (edit_num=2048) across Reliability, Generality, Locality, and overall score.

| Model | Method | Reliability ↑ | | Generality ↑ | | Locality ↑ | | Score ↑ |
|---|---|---|---|---|---|---|---|---|
| | | AG | TF | AG | TF | AG | TF | |
| Phi-2 | FT | 97.41 | 98.42 | **77.68** | **86.60** | 8.54 | 51.49 | 33.08 |
| | MEMIT | 91.89 | 94.94 | 64.01 | 79.90 | 20.02 | 68.03 | 52.54 |
| | PMET | 29.79 | 56.67 | 18.75 | 47.84 | 22.12 | 69.48 | 32.42 |
| | MALMEN | 99.36 | **99.71** | 72.80 | 86.31 | 18.01 | 78.90 | 52.81 |
| | SCEN | 79.29 | 85.43 | 47.70 | 62.72 | 90.96 | 93.13 | 72.33 |
| | **MeG (ours)** | **99.61** | 99.42 | 76.37 | 84.27 | **91.21** | **96.33** | **90.36** |
| GPT-J | FT | **100.0** | 99.83 | **100.0** | **99.83** | 4.34 | 41.92 | 20.39 |
| | MEMIT | 96.19 | 98.12 | 75.05 | 84.91 | 35.16 | 82.27 | 69.54 |
| | PMET | 84.72 | 92.36 | 70.46 | 83.28 | 29.10 | 78.67 | 62.56 |
| | MALMEN | 99.90 | **99.98** | 79.68 | 89.61 | 17.87 | 82.82 | 53.69 |
| | SCEN | 89.74 | 93.11 | 42.87 | 57.51 | 62.50 | 86.40 | 66.54 |
| | **MeG (ours)** | 99.32 | 99.34 | 82.71 | 88.09 | **81.25** | **92.43** | **89.95** |
| Llama-3 | FT | **100.00** | 99.90 | 52.88 | **91.51** | 11.87 | 53.93 | 39.31 |
| | MEMIT | 94.78 | 97.33 | **77.73** | 86.83 | 49.90 | 84.78 | 77.88 |
| | PMET | 80.27 | 88.94 | 70.56 | 82.57 | 45.61 | 82.06 | 71.35 |
| | MALMEN | 99.26 | 99.72 | 74.60 | 88.50 | 47.65 | 88.50 | 77.83 |
| | **MeG (ours)** | 99.90 | 99.83 | 76.81 | 86.82 | **84.03** | **94.48** | **89.49** |

Table 11: Performance comparison on Phi-2, GPT-J, and Llama-3 models for ZsRE dataset (edit_num=4096) across Reliability, Generality, Locality, and overall score.

| Model | Method | Reliability ↑ | | Generality ↑ | | Locality ↑ | | Score ↑ |
|---|---|---|---|---|---|---|---|---|
| | | AG | TF | AG | TF | AG | TF | |
| **Phi-2** | FT | 96.67 | 98.36 | **76.24** | **86.23** | 9.91 | 52.55 | 36.33 |
| | MEMIT | 86.33 | 92.16 | 57.47 | 75.40 | 13.57 | 61.02 | 41.91 |
| | PMET | 25.10 | 52.51 | 16.60 | 45.11 | 19.17 | 66.26 | 28.77 |
| | MALMEN | 97.70 | 98.88 | 66.99 | 83.02 | 16.84 | 78.04 | 50.20 |
| | **MeG (ours)** | **98.93** | **99.10** | 68.92 | 79.76 | **92.70** | **96.73** | **87.76** |
| **GPT-J** | FT | **100.0** | 99.84 | **100.0** | 99.84 | 4.68 | 41.31 | 21.59 |
| | MEMIT | 93.70 | 97.03 | 68.41 | 82.13 | 29.52 | 78.97 | 63.62 |
| | PMET | 82.32 | 90.91 | 63.43 | 78.85 | 24.71 | 74.65 | 56.89 |
| | MALMEN | 99.92 | **99.98** | 76.29 | 87.66 | 15.38 | 78.87 | 49.09 |
| | **MeG (ours)** | 98.93 | 99.15 | 70.80 | 80.27 | **83.47** | **93.38** | **86.37** |
| **Llama-3** | FT | **100.0** | **99.91** | 56.25 | **90.32** | 13.31 | 53.80 | 42.08 |
| | MEMIT | 92.80 | 96.38 | 71.85 | 84.19 | 37.13 | 77.22 | 69.10 |
| | PMET | 75.17 | 85.42 | 64.92 | 79.12 | 36.94 | 76.31 | 64.36 |
| | MALMEN | 98.68 | 99.65 | 73.29 | 88.69 | 46.02 | 86.91 | 76.60 |
| | **MeG (ours)** | 99.88 | 99.83 | **73.58** | 85.15 | **81.62** | **93.21** | **87.79** |

Table 12: Performance comparison on Phi-2, GPT-J, and Llama-3 models for COUNTERFACT dataset (edit_num=1024) across Reliability, Generality, Locality, and overall score.

| Model | Method | Reliability ↑ | | Generality ↑ | | Locality ↑ | | Score ↑ |
|---|---|---|---|---|---|---|---|---|
| | | AG | TF | AG | TF | AG | TF | |
| **Phi-2** | FT | 99.02 | 99.02 | 40.82 | 41.02 | 1.46 | 15.84 | 7.34 |
| | MEMIT | 99.61 | 99.61 | 49.90 | 49.71 | 17.58 | 63.25 | 45.14 |
| | PMET | 68.95 | 68.75 | 29.88 | 29.69 | 25.98 | 71.44 | 40.35 |
| | MALMEN | 99.70 | 99.60 | 35.64 | 35.54 | 9.86 | 45.46 | 30.05 |
| | SCEN | 98.73 | 98.73 | 3.42 | 3.42 | 77.05 | 82.82 | 9.52 |
| | **MeG (ours)** | **99.90** | **99.90** | **61.82** | **66.11** | **78.71** | **86.27** | **79.35** |
| **GPT-J** | FT | **100.0** | **100.0** | 33.69 | 33.79 | 4.39 | 13.51 | 15.74 |
| | MEMIT | 99.71 | 99.71 | 67.58 | 67.38 | **51.86** | **70.37** | **72.12** |
| | PMET | 99.41 | 99.41 | 75.20 | 75.00 | 45.41 | 64.15 | 71.12 |
| | MALMEN | 99.70 | 99.70 | 37.30 | 37.59 | 24.60 | 41.30 | 43.37 |
| | SCEN | 91.40 | 91.40 | 14.55 | 14.65 | 29.10 | 52.34 | 28.26 |
| | **MeG (ours)** | **100.0** | **100.0** | **81.64** | **82.91** | 44.53 | 51.70 | 69.68 |
| **Llama-3** | FT | **100.0** | **100.0** | 18.07 | 18.65 | 1.17 | 18.23 | 5.78 |
| | MEMIT | 99.61 | 99.61 | 66.99 | 66.65 | 44.24 | **77.22** | 70.12 |
| | PMET | 98.14 | 98.14 | 64.45 | 64.21 | 40.14 | 73.49 | 66.67 |
| | MALMEN | 98.34 | 98.14 | 40.23 | 40.77 | 14.45 | 44.24 | 37.14 |
| | **MeG (ours)** | 99.61 | 99.61 | **68.07** | **69.73** | **57.52** | 69.51 | **74.18** |

Table 13: Performance comparison on Phi-2, GPT-J, and Llama-3 models for COUNTERFACT dataset (edit_num=2048) across Reliability, Generality, Locality, and overall score.

| Model | Method | Reliability ↑ | | Generality ↑ | | Locality ↑ | | Score ↑ |
|-------|--------|------|------|------|------|------|------|------|
| | | AG | TF | AG | TF | AG | TF | |
| | FT | 98.53 | 98.53 | 37.50 | 37.55 | 1.02 | 16.85 | 5.39 |
| | MEMIT | **99.07** | **99.02** | 48.68 | 48.39 | 16.31 | 59.17 | 42.98 |
| Phi-2 | PMET | 66.65 | 66.36 | 27.05 | 26.90 | 26.56 | 70.26 | 38.44 |
| | MALMEN | 98.58 | 98.58 | 31.44 | 31.68 | 4.68 | 32.96 | 18.31 |
| | SCEN | 75.97 | 76.02 | 2.64 | 2.59 | 63.86 | 82.59 | 7.33 |
| | **MeG (ours)** | 98.58 | 98.58 | **60.06** | **63.38** | **73.78** | **83.12** | **76.63** |
| | FT | **100.0** | **100.0** | 41.99 | 42.38 | 2.53 | 12.05 | 11.00 |
| | MEMIT | 99.27 | 99.22 | 61.72 | 61.52 | 43.65 | 63.97 | 65.82 |
| GPT-J | PMET | 98.88 | 98.83 | 74.41 | 74.37 | 40.43 | 61.06 | 68.01 |
| | MALMEN | 99.75 | 99.80 | 33.20 | 33.30 | 25.53 | 40.62 | 41.67 |
| | SCEN | 85.15 | 85.15 | 13.28 | 13.53 | 30.22 | 52.85 | 26.70 |
| | **MeG (ours)** | 99.95 | 99.95 | **78.08** | **79.10** | 62.94 | 66.37 | **78.52** |
| | FT | **100.0** | **100.0** | 23.05 | 23.80 | 0.98 | 16.81 | 5.06 |
| | MEMIT | 99.07 | 99.02 | **63.13** | 63.26 | **63.26** | 70.25 | **73.27** |
| Llama-3 | PMET | 96.88 | 96.70 | 62.35 | 62.45 | 32.76 | 66.41 | 61.04 |
| | MALMEN | 93.46 | 93.51 | 32.03 | 32.69 | 13.23 | 42.31 | 32.89 |
| | **MeG (ours)** | 99.17 | 99.17 | 60.69 | **63.60** | 60.25 | **72.59** | 72.52 |

Table 14: Performance comparison on Phi-2, GPT-J, and Llama-3 models for COUNTERFACT dataset (edit_num=4096) across Reliability, Generality, Locality, and overall score.

| Model | Method | Reliability ↑ | | Generality ↑ | | Locality ↑ | | Score ↑ |
|-------|--------|------|------|------|------|------|------|------|
| | | AG | TF | AG | TF | AG | TF | |
| | FT | 97.33 | 97.33 | 38.55 | 38.55 | 0.95 | 16.54 | 5.06 |
| | MEMIT | 97.63 | 97.61 | 46.39 | 46.14 | 12.38 | 52.38 | 36.68 |
| Phi-2 | PMET | 62.16 | 61.77 | 26.00 | 25.81 | 22.97 | 66.31 | 35.69 |
| | MALMEN | 97.70 | 97.60 | 27.95 | 28.10 | 5.90 | 38.57 | 20.89 |
| | **MeG (ours)** | **98.41** | **98.41** | **56.15** | **59.72** | **73.56** | **82.25** | **74.41** |
| | FT | **100.0** | **100.0** | 48.43 | 48.49 | 1.92 | 9.73 | 8.76 |
| | MEMIT | 98.34 | 98.34 | 61.60 | 61.45 | 33.08 | 58.18 | 59.84 |
| GPT-J | PMET | 98.63 | 98.63 | **71.12** | **71.00** | 33.42 | 54.74 | 62.10 |
| | MALMEN | 99.97 | 99.97 | 32.12 | 32.15 | 17.30 | 32.13 | 35.05 |
| | **MeG (ours)** | 99.34 | 99.34 | 68.99 | 69.95 | **65.48** | **67.54** | **75.95** |
| | FT | **100.0** | **100.0** | 26.78 | 27.50 | 0.61 | 15.22 | 3.34 |
| | MEMIT | 98.95 | 98.85 | **61.06** | **61.19** | 16.75 | 54.93 | 45.85 |
| Llama-3 | PMET | 95.24 | 95.18 | 60.77 | 60.68 | 24.98 | 58.28 | 53.99 |
| | MALMEN | 97.51 | 97.46 | 32.89 | 33.62 | 10.55 | 39.09 | 29.85 |
| | **MeG (ours)** | 98.73 | 98.75 | 55.42 | 58.42 | **61.28** | **72.61** | **70.17** |

## A.3 FURTHER ENHANCING LOCALITY VIA DIFFUSION

In addition to improving the Locality metric through our MeG's Familiarity Network and single-neuron addition design, we can further enhance Locality by leveraging the property of our weight generation model. Specifically, we guide the diffusion model to generate all-zero weights for irrelevant knowledge queries, thereby preserving Locality. To achieve this, we collect a set of pseudo-irrelevant knowledge (distinct from the edited knowledge) and construct pseudo-irrelevant knowledge–zero weight pairs, which are incorporated into the diffusion model's training data. The experimental results, recorded in Table 15, demonstrate that this Data Augmentation (DA) further improves Locality performance. However, it also leads to a moderate decline in Generality. Depending on the practical application requirements, users can flexibly choose between these trade-offs.

Table 15: Performance comparison on Phi-2 (2.7B) and GPT-J (6B) models for ZsRE and COUNTERFACT (edit_num=1024) across Reliability, Generality, Locality, and overall Score between our MeG and it with Data Augmentation for the diffusion model component.

| Dataset | Model | Method | Reliability ↑ | | Generality ↑ | | Locality ↑ | | Score ↑ |
|---|---|---|---|---|---|---|---|---|---|
| | | | AG | TF | AG | TF | AG | TF | |
| ZsRE | Phi-2 | MeG | **99.61** | **99.49** | **80.37** | **87.24** | 88.67 | 95.75 | 91.30 |
| | | MeG+DA | 99.51 | 99.39 | 78.52 | 85.72 | **97.36** | **97.97** | **92.32** |
| | GPT-J | MeG | 99.41 | 99.25 | **83.59** | **89.09** | 75.29 | 90.35 | 88.66 |
| | | MeG+DA | 99.41 | 99.25 | 81.54 | 86.99 | **96.58** | **97.41** | **93.00** |
| COUNTERFACT | Phi-2 | MeG | **99.90** | **99.90** | **61.82** | **66.11** | 78.71 | 86.27 | 79.35 |
| | | MeG+DA | 98.83 | 98.83 | 58.40 | 62.30 | **89.55** | **92.74** | **79.62** |
| | GPT-J | MeG | **100.00** | **100.00** | **81.64** | **82.91** | 44.53 | 51.70 | 69.68 |
| | | MeG+DA | 99.51 | 99.51 | 75.20 | 76.37 | **86.72** | **83.72** | **85.76** |

## A.4 LAYER SELECTION FOR NEURON ADDING

Our MeG operates by adding one generated neuron at one FFN layer of LLMs. We observe that the Reliability performance varies depending on which FFN layer the neuron is added to. To investigate this, we test the Reliability of editing on the ZsRE and COUNTERFACT datasets by adding a single neuron to different FFN layers in Phi-2, GPT-J and Llama-3, and the Reliability under AG generation setting is in Fig. 5, Fig. 6 and Fig. 7 respectively. From the figure, we can find that the Reliability performance differs markedly across layers but exhibits cross-dataset consistency for each LLM. Based on these findings, our MeG selects the FFN block in layer 29 for Phi-2, layer 9 for GPT-J, and layer 14 for Llama-3 as the optimal layer for dynamic neuron adding. Besides, from the SR performance of applying neuron edits at different layers of the three LLMs, it can be observed that as the model size increases, the "sweet spot" for selecting layers becomes broader, with more layers

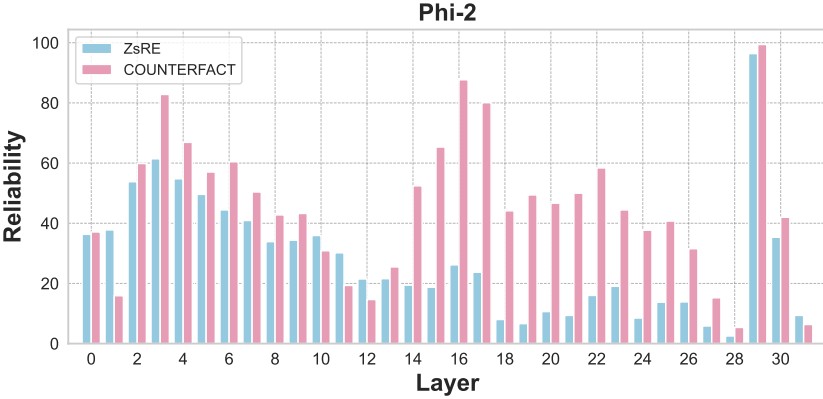

Figure 5: Reliability performance with different FFN layers selected for adding a single neuron of Phi-2 (2.7B).

exhibiting excellent performance. This demonstrates the strong scalability of our method with LLM sizes.

## A.5 How to Select the Threshold of Our Familiarity Network

In this section, we explain how to select the entropy threshold hyperparameter ($\epsilon$) for the Familiarity Network in MeG. We determine $\epsilon$ via grid search to balance Generality and Locality. The Familiarity Network is trained to overfit edited knowledge, assigning it a "low-entropy fingerprint". Consequently, the entropy distribution follows a pattern: Edited Queries (Lowest) ¡ Paraphrases (Slightly Higher) ¡¡ Irrelevant Knowledge (Highest). And there is a small overlap between the entropy of paraphrases and irrelevant knowledge. We select $\epsilon$ to separate these two groups optimally. We conducted a sensitivity analysis of the entropy threshold for Phi-2 on ZsRE at 1024 KE scale(see the Table 16).

In Table 16, since the entropy of the original edited knowledge output is very low, Reliability remains almost unaffected by changes in the entropy threshold. The entropy of the paraphrases is slightly higher than that of the original edited knowledge. If the entropy threshold is set too low, some paraphrases may be misclassified as irrelevant knowledge, thereby affecting the Generality metric. As the entropy threshold increases, the number of misclassified paraphrases decreases, leading to an improvement in Generality. However, because the entropy of paraphrases does not reach excessively high values, Generality eventually saturates as the entropy threshold continues to increase.

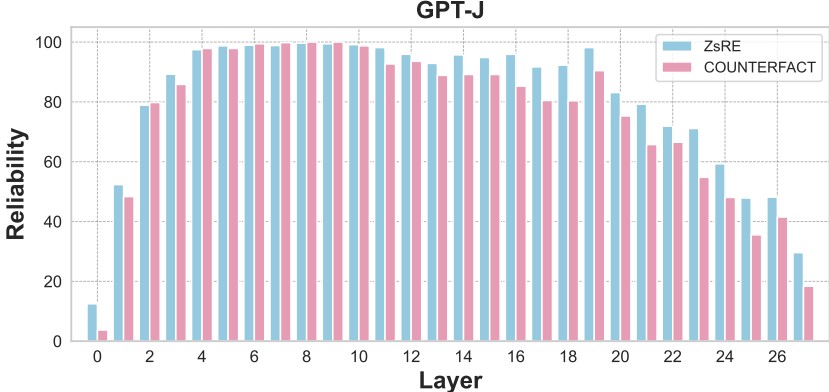

Figure 6: Reliability performance with different FFN layers selected for adding a single neuron of GPT-J (6B).

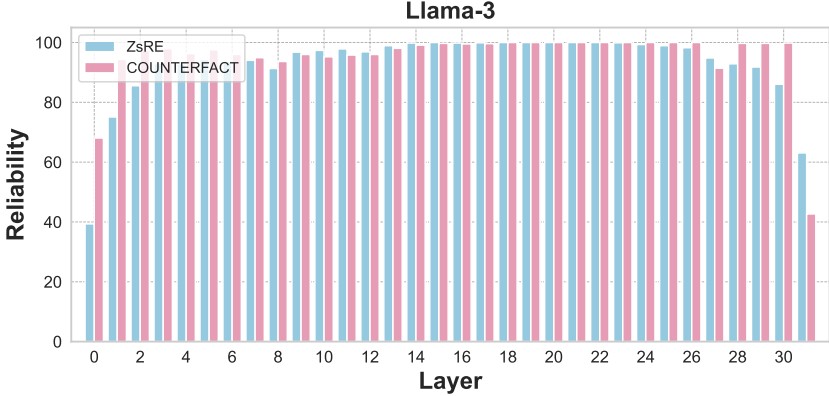

Figure 7: Reliability performance with different FFN layers selected for adding a single neuron of Llama-3 (8B).

Table 16: Sensitivity of our method to different entropy thresholds of Familiarity Network for Phi-2 on ZsRE (edit_num=1024) across Reliability, Generality, Locality.

| Threshold | Reliability ↑ | | Generality ↑ | | Locality ↑ | |
|---|---|---|---|---|---|---|
| | AG | TF | AG | TF | AG | TF |
| **0.05** | 99.61 | 99.49 | 78.71 | 86.12 | 92.09 | 96.97 |
| **0.1** | 99.61 | 99.49 | 80.37 | 87.24 | 89.67 | 95.75 |
| **0.15** | 99.80 | 99.61 | 81.15 | 87.68 | 86.72 | 94.82 |
| **0.2** | 99.80 | 99.61 | 81.54 | 88.14 | 84.67 | 94.05 |
| **0.25** | 99.80 | 99.61 | 82.03 | 88.13 | 82.23 | 93.36 |
| **0.3** | 99.80 | 99.61 | 82.13 | 88.32 | 81.15 | 92.64 |

Irrelevant knowledge generally has a higher entropy than both the original edited knowledge and paraphrases. When the entropy threshold is low, the vast majority of irrelevant knowledge is correctly classified. However, as the entropy threshold increases, a portion of irrelevant knowledge is gradually misclassified as edited knowledge, causing a decline in Locality performance. Locality continues to decrease as the entropy threshold increases.

Overall, the changes in Generality and Locality with respect to the entropy threshold are gradual and do not exhibit dramatic fluctuations. To achieve a better balance between Generality and Locality, we set the entropy threshold here to 0.1.

## A.6 EDITING CAPACITY OF SINGLE ADDED NEURON

Since our MeG approach utilizes the addition of a single neuron for knowledge editing, a reasonable concern is that the "editing capacity" associated with the weight of a single neuron may be insufficient to accomplish the editing task in certain scenarios (e.g., when the knowledge text to be edited is relatively long), which could severely impact the reliability performance of the method. To address this concern, our conclusion is that in the vast majority of text-based knowledge editing scenarios, the addition of a single neuron is sufficient to fulfill the required knowledge editing. We will discuss this from both theoretical and experimental perspectives.

### A.6.1 SINGLE ADDED NEURON FOR LONG-TEXT KNOWLEDGE EDITING

Generally speaking, a single neuron is sufficient to encode various knowledge updates. Theoretically, the generated associative weights for a single neuron are two vectors with thousands of dimensions. The generated neuron does not need to "store" the entire knowledge fact from scratch. Instead, it acts as a "steering vector" or a "linear probe" that intervenes in the hidden states of the pre-trained LLM. Recent studies on the Linear Representation Hypothesis (Park et al., 2024; Zou et al., 2025) suggest that concepts and knowledge in LLMs are often encoded linearly. Therefore, a single neuron is usually sufficient to shift the activation trajectory from the "old fact" to the "new fact" by leveraging the model's existing computational capacity.

To validate this empirically, we conducted additional experiments to demonstrate that the capacity of a single neuron is sufficient. In this experiment, we specifically randomly select a subset with 100 cases from the long-sequence text generation knowledge UnKEBench (Deng et al., 2025), where each piece of knowledge requires the generation of approximately 100 tokens (The number of tokens that most knowledge editing tasks need to generate is far less than 100). We use only a single additional neuron attached to 3 different LLMs, and only allow the weight of this neuron to be fine-tuned. We

Table 17: Performance of the accuracy while fine-tuning a single added neuron within LLMs in a randomly selected subset of UnKEBench. "FT_ACC" means "Fine-tuning Accuracy".

| LLM | FT_ACC ↑ |
|---|---|
| Phi-2 | 98.61 |
| GPT-J | 100.00 |
| Llama-3 | 100.00 |

recorded the fine-tuning accuracy of one single neuron on each LLM in Table 17. We followed (Deng et al., 2025) and employed semantic-accuracy-based metrics (BERT Score) to evaluate the editing precision for long-text knowledge.

Table 18: Performance of Phi-2 utilizes our MeG on MQuAKE-CF with 1024 editing scale.

| Method | Edit-wise ↑ | Instance-wise ↑ | Multi-hop ↑ |
|--------|-------------|-----------------|-------------|
| MEMIT | 79.30 | 62.01 | 0.10 |
| MALMEN | 90.97 | 82.52 | 5.66 |
| **MeG(ours)** | **98.24** | **96.88** | **41.41** |

Experimental results in Table 17 show that in Unke, the fine-tuning accuracy of one single added neuron can be achieved close to 100 across all three LLMs. It indicates that the editing capacity of a single neuron is sufficient for typical knowledge editing tasks.

### A.6.2 SINGLE ADDED NEURON FOR MULTI-HOP REASONING KNOWLEDGE EDITING

To evaluate whether our method can generalize beyond paraphrases to more complex multi-hop reasoning KE data, we have experimentally evaluated the performance of our MeG for Phi-2 on the MQuAKE-CF dataset from the multi-hop reasoning KE dataset, MQuAKE (Zhong et al., 2023), when editing 1024 instances. In our experiment, a single neuron is collected to correspond to multiple pieces of atomic knowledge within each editing instance (the answers to multi-hop questions are derived from the combination of these atomic pieces of knowledge). Approximately 2,000 instances from MQuAKE-CF, excluding the 1024 selected for editing, have been collected as pseudo-editing data. We have trained the Text Encoder, BERT, using contrastive learning based on InfoNCE Loss to align the representations of multi-hop questions and their atomic knowledge within the pseudo-editing data. We hope this would generalize to real editing instances, bringing the representations of multi-hop questions closer to their corresponding atomic knowledge, thereby allowing DiT to generate neurons near those corresponding to the atomic knowledge.

We only use the atomic knowledge and their corresponding neuron collected for them in the editing instances to train DiT, aiming to evaluate the generalization ability of our method to multi-hop questions after editing atomic knowledge. Additionally, we have conducted experiments with 2 baseline massive-editing methods, MEMIT (representing the locate-then-edit methods) and MAL-MEN (representing the hypernetwork methods), with the same editing setting as a comparison. The experimental results are recorded in Table 18. (The evaluation metric is aligned with (Zhong et al., 2023).)

From Table 18, it can be observed that our MeG method, utilizing an additional single-neuron attachment, generalizes well to complex multi-question-answering tasks. It demonstrates the robust performance of our method in both atomic knowledge editing and multi-hop task performance.

### A.7 COMPARISON WITH MORE METHODS

In this section, we will discuss in detail whether and how a comparison between our MeG with two sequential editing methods, AlphaEdit (Fang et al., 2025) and RLEdit (Li et al., 2025) can be conducted.

### A.7.1 OUR MEG VS. ALPHAEDIT

Although the experiments in the AlphaEdit paper are primarily based on a sequential editing setup, the method itself, as well as its engineering resource requirements, supports one-batch massive editing. Therefore, we conducted knowledge editing experiments on AlphaEdit using the one-batch editing setup. We have conducted the experiments of KE as the editing scale range from 1024 to 10K, for Phi-2 on ZsRE utilizing AlphaEdit. Comparison Results of AlphaEdit and our MeG are recorded in Table 19.

From the experimental results in the table above, it can be observed that our MeG method demonstrates a clear advantage over the AlphaEdit method across the three knowledge editing performance metrics under both Prefix-autoregressive (AG) and Teacher Forcing (TF) generation settings, especially in terms of Locality performance. This highlights the superiority of our method compared to AlphaEdit in large-scale knowledge editing tasks.

Table 19: Performance comparison between our MeG vs AlphaEdit under massive-scale batch editing for Phi-2 on ZsRE (under various editing scales) across Reliability, Generality, Locality, and overall Score.

| Edit_Scale | Method | Reliability ↑ | | Generality ↑ | | Locality ↑ | | Score ↑ |
|---|---|---|---|---|---|---|---|---|
| | | AG | TF | AG | TF | AG | TF | |
| 1024 | AlphaEdit | 97.36 | 99.26 | 64.45 | 81.83 | 24.41 | 69.89 | 58.05 |
| | MeG(Ours) | **99.61** | **99.49** | **80.37** | **87.24** | **88.67** | **95.75** | **91.30** |
| 2048 | AlphaEdit | 95.21 | 98.35 | 61.52 | 79.62 | 16.70 | 64.93 | 48.09 |
| | MeG(Ours) | **99.61** | **99.42** | **76.37** | **84.27** | **91.21** | **96.33** | **90.36** |
| 4096 | AlphaEdit | 92.70 | 97.35 | 61.11 | 79.13 | 16.04 | 62.57 | 46.73 |
| | MeG(Ours) | **98.93** | **99.10** | **68.92** | **79.76** | **92.70** | **96.73** | **87.76** |
| 10000 | AlphaEdit | 72.43 | 86.74 | 40.27 | 64.69 | 9.87 | 56.32 | 32.49 |
| | MeG(Ours) | **95.07** | **97.04** | **59.69** | **74.17** | **91.14** | **95.84** | **82.80** |

Next, we attempt to theoretically explain the reasons behind the advantages of our MeG method over the AlphaEdit method in large-scale batch knowledge editing scenarios.

AlphaEdit is a locate-then-edit type of knowledge editing method that optimizes how this approach modifies the internal weights of LLMs to edit new knowledge without forgetting unrelated knowledge. Specifically, it projects weight modifications into the null space of unrelated knowledge, thereby preserving unrelated knowledge and avoiding the difficulty of balancing issues faced by earlier methods that directly used weighted error losses between old and new knowledge. **However, AlphaEdit still faces two challenges.**

**First**, constructing the null space of unrelated knowledge requires a set of unrelated knowledge. Given that LLMs store an enormous amount of unrelated knowledge, this set is extremely large and almost impossible to fully capture. Currently, locate-then-edit methods like AlphaEdit and MEMIT approximate this set by randomly sampling 100k Wikipedia entries. However, this approximation is far from sufficient, making it difficult to obtain an accurate null space of unrelated knowledge, which in turn affects the locality performance of such knowledge editing methods. As discussed in our paper, as the scale of editing increases, the interference of those methods with the original LLM grows, and the locality metric continues to decline (as shown in the experimental results in the table above).

In contrast, our MeG method employs a familiarity network mechanism that does not require any unrelated knowledge. Instead, it uses the much smaller and more accessible set of editing knowledge to train the familiarity network to overfit to the assigned random "low-entropy fingerprints" of these editing knowledge entries. Based on statistical theory and the properties of neural networks, any unrelated knowledge not seen during training will, with high probability, output a high-entropy distribution when passed through the familiarity network. By using an entropy threshold, unrelated knowledge can be effectively and accurately distinguished from editing knowledge and equivalent expressions. This establishes a low-cost and highly accurate routing mechanism for old and new knowledge.

Additionally, we have an extra protection for Locality. For the very few unrelated knowledge queries that are incorrectly routed by the familiarity network, the single neuron generated by DiT will likely have limited interference with the original LLM, thus not affecting its responses to unrelated knowledge. As shown in the table above, our method demonstrates a significant advantage over AlphaEdit in terms of Locality performance, and this advantage grows as the scale of editing increases.

**The second potential issue AlphaEdit faces** is that it constrains weight updates within the null space of unrelated knowledge. As the scale of editing increases, this may lead to insufficient null space capacity or overly restrictive constraints, thereby affecting its ability to update editing knowledge. From the table above, it can be observed that when the editing scale increases to 10,000, AlphaEdit's Reliability metric drops significantly. In contrast, our method leverages DiT to dynamically generate neuron weights for new knowledge updates, offering a much larger editing capacity. Our experiments also show that even at an editing scale of 10,000, the Reliability metric of our method remains at a very high level.

Table 20: Training time (hours) of Phi-2 utilizing our MeG within 1024 editing scale on ZsRE.

| Scale | Weight Collection | DiT Training | Total |
|---|---|---|---|
| **1024** | 0.114 | 2.980 (1 GPU) | 3.094 |
| **2048** | 0.232 | 4.237 | 4.469 |
| **4096** | 0.469 | 7.929 | 8.398 |
| **10000** | 1.175 | 11.66 | 12.835 |

Table 21: Inference throughput (QPS) of various LLMs after editing using different KE methods.

| LLM | MEMIT | PMET | MALMEN | MeG(Ours) |
|---|---|---|---|---|
| **Phi-2** ↑ | 2.38 | 2.48 | 1.38 | **6.82** |
| **GPT-J** ↑ | 1.32 | 1.29 | 0.85 | **2.75** |
| **Llama-3** ↑ | 1.55 | 1.54 | 1.05 | **3.70** |

### A.7.2 OUR MEG VS. RLEDIT

RLEdit focuses on the issue of performance degradation in knowledge editing caused by LLM weight parameter drift in sequential editing settings for hypernetwork-based methods. Its focus is on sequential editing problems, and experiments show that as the editing scale increases, the method requires a large amount of GPU memory, making it difficult to apply to one-batch massive editing. Therefore, we did not include it in experimental comparisons.

### A.8 FURTHER EFFICIENCY ANALYSIS

In this section, we will introduce a further efficiency analysis of our MeG method, including training time and inference throughput analysis.

### A.8.1 TRAINING TIME

Table 20 lists the training time costs (hours) of our MeG for Phi-2-ZsRE KE at different editing scales (including weight collection and diffusion model training). The hardware used for training DiT in the 1024 KE scale is 1 NVIDIA RTX 4090D GPU, and for other scales consists of 4 NVIDIA RTX 4090D GPUs.

Table 20 lists the training time costs. Considering the significant improvements in editing performance, the fact that the training cost is a one-time investment, and, more importantly, the efficiency during the inference phase, we believe this time cost is acceptable.

### A.8.2 INFERENCE EFFICIENCY

To compare the inference efficiency of different knowledge editing methods, we have tested the inference throughput (the number of queries LLMs respond to per second, **QPS**) after applying different knowledge editing methods on three LLMs, and the results are recorded in Table 21.

Our experiments show that **our MeG achieves higher practical throughput (QPS)** than baselines. This is attributed to **three factors**:

**DDIM Acceleration**: As it is discussed in Sec. 6.3 and shown in Table 8, we employ DDIM sampling to accelerate the diffusion process (reduce the sampling steps from 1000 to 50). The overhead of generating dynamic weights is marginal, adding only 5% to the total inference time compared to a standard forward pass.

**Selective Routing**: The Familiarity Network filters out irrelevant queries, allowing them to bypass the diffusion module entirely, incurring zero overhead for general knowledge.

**Generation Efficiency (Crucial Observation)**: Most importantly, baselines suffer from severe performance degradation at large-scale KE, often falling into repetition loops or generating excessively long, nonsensical tokens. This drastically increases their inference time per query. In contrast, MeG maintains high performance and generates concise, correct answers.

As discussed above, during the inference phase, our method demonstrates higher practical system efficiency compared to the baseline methods.

## A.9 DATASET PRE-PROCESSING

For ZsRE, we construct our experimental data by combining the test set and a subset of 10,000 samples from the training set provided by (Yao et al., 2023b), resulting in a total of 29,086 samples. Notably, we observe that some entries contain mutually equivalent factual representations, which may lead to inflated performance during evaluation—particularly on the Generality metric—due to redundancy. To mitigate this issue, we retain only one instance per subject entity, thereby eliminating overlapping facts that could potentially influence each other. This filtering yields a refined ZsRE dataset containing 19,964 unique samples. Furthermore, given the suboptimal quality of the original equivalent representations, we systematically recollect 10 equivalent representations for each edited instance generated by GLM-4-9B-Chat (GLM et al., 2024) with Table 22. Additionally, we strictly retain only instances for which baseline models produce incorrect predictions, ensuring that our error correction experiments focused exclusively on model mistakes. For the COUNTERFACT dataset, we use the same data split as MEMIT (Meng et al., 2023) with 20,877 items, which applies a conflict filtering step to remove contradictory instances from the original COUNTERFACT dataset. For both datasets, we additionally collect pre-edit model responses for a set of predefined queries before performing any edits, enabling us to assess locality, that the degree to which updates are confined to the targeted knowledge without affecting unrelated facts.

## A.10 DETAILS OF GENERAL BENCHMARKS

**BBH** (Suzgun et al., 2023) includes 6.5K challenging problems across 23 tasks from curated subset of BIG-Bench evaluation, designed to assess language models' ability to solve complex multi-step reasoning problems in domains like logical deduction, commonsense understanding, and algorithmic manipulation, with explicit analysis of chain-of-thought prompting efficacy.

**GSM8K** (Cobbe et al., 2021) is a high-quality dataset of 8.5K grade-school level math word problems, designed to evaluate and train models on step-by-step mathematical reasoning.

**MMLU** (Hendrycks et al.) is a multidisciplinary benchmark comprising 57 subjects, designed to comprehensively evaluate language models' breadth of knowledge and reasoning capabilities across diverse domains, including STEM, humanities, and social sciences, through multiple-choice question answering.

**RTE** (Dagan et al., 2005) consists of sentence pairs annotated for whether the hypothesis is entailed by the premise, serving as a benchmark for evaluating textual entailment ability.

**SST-2** (Socher et al., 2013) is a binary sentiment classification dataset derived from the Stanford Sentiment Treebank, consisting of movie review sentences labeled as either positive or negative.

## A.11 MORE IMPLEMENTATION DETAILS

### A.11.1 MORE DIFFUSION MODEL IMPLEMENTATION DETAILS OF OUR MEG

We implement our diffusion model based on DiT, following the default DiT-B/2 configuration across all models and datasets. Unlike the original DiT framework, we introduce several modifications. In the input layer, instead of mapping inputs to a latent space, we directly feed in raw weight parameters. Specifically, For Phi-2 neurons, the input shape is `(batch_size, 5121)`, for GPT-J neurons, it is `(batch_size, 8193)`, and for LLama-3 neurons, it is `(batch_size, 8192)`. These inputs are then divided into non-overlapping patches with a patch size of 100. For conditioning, we directly use the output of BERT models that are fine-tuned on tasks corresponding to each specific model and dataset, and inject them via the `[CLS]` token. In the output layer, we ensure the dimensional consistency to align with the original input shape.

During training, we use the AdamW optimizer with a fixed learning rate of $1 \times 10^{-4}$. We evaluate the generation performance every 1000 epochs. If the L2 norm between the generated parameters and the target weights yields a mean plus standard deviation (mean + std) below 0.5, we consider the training to be successful.

Table 22: Prompt used for rephrases generation.

| Prompt used for rephrases generation. |
| --- |

You are a professional linguist. Please give 11 different rephrases of the sentence apart from itself. Please provide your answers directly within <list ></list>, without including any additional content.

sentence: Who was the designer of Tor missile system?
rephrases: <list>["The designer of Tor missile system was who?", "Who designed the Tor missile system?", "What's the name of the person who designed the Tor missile system?", "Who worked on designing Tor missile system?", "The Tor missile system was designed by whom?", "Who was responsible for designing the Tor missile system?", "The Tor missile system's designer is who?", "Who is credited with designing the Tor missile system?", "By whom was the Tor missile system designed?", "Whom do we credit as the designer of the Tor missile system?", "The Tor missile system's design is attributed to whom?"]</list>

sentence: When was USA Trains established?
rephrases: <list>["When was USA Trains founded?", "When did USA Trains start?", "When was USA Trains created?", "When did USA Trains come into existence?", "USA Trains was established in which year?", "When did USA Trains start operating?", "What is the inception date of USA Trains?", "In which year was USA Trains established?", "When did USA Trains begin?", "At what time was USA Trains founded?", "What was the establishment date of USA Trains?"]</list>

sentence: What artist released Bitter Heart?
rephrases:<list>["Which artist released the song 'Bitter Heart'?", "Who is the artist behind 'Bitter Heart'?", "The song 'Bitter Heart' was released by which artist?", "Who performed 'Bitter Heart'?", "Who sang 'Bitter Heart'?", "The artist who released 'Bitter Heart' is who?", "Who is credited with releasing 'Bitter Heart'?", "Bitter Heart was first performed by which artist?", "Who is the artist that released 'Bitter Heart'?", "Whom do we attribute as the artist of 'Bitter Heart'?", "The release of 'Bitter Heart' is attributed to whom?"]</list>

sentence: {QUESTION}
rephrases:

Table 23: Prompts for testing GSM8K, RTE and SST-2.

| Prompts for testing GSM8K, RTE and SST-2. |
| --- |
| **GSM8K**
*Phi-2:* Question: {QUESTION} Output: Let's think step by step.
*GPT-J: Question:* {QUESTION} Answer: Let's think step by step.

**RTE**
*Phi-2:* Instruct: {SENTENCE1} entails the {SENTENCE2}. True or False? Output:
*GPT-J:* {SENTENCE1} entails the {SENTENCE2}. True or False? answer:

**SST-2**
*Both Phi-2 and GPT-J:* For each snippet of text, label the sentiment of the text as positive or negative. The answer should be exact 'positive' or 'negative'. text: {TEXT} answer: |

To improve training efficiency and stability, we adopt the velocity prediction formulation (v-prediction) in our diffusion model. Additionally, we employ mixed-precision training following the implementation from https://github.com/chuanyangjin/fast-DiT.

All training tasks are conducted on NVIDIA RTX 4090(or 4090D) GPUs to facilitate consistent comparison of convergence performance across different models and datasets.

### A.11.2 PROMPTS OF LLMS FOR TESTING ON GENERAL BENCHMARKS

The prompts for testing GSM8K, RTE and SST-2 are shown in Table 23.

The prompts for evaluating BBH and MMLU are following the code released in https://github.com/FranxYao/chain-of-thought-hub/tree/main.

### A.12 REPRODUCTION DETAILS OF BASELINE METHODS

**Fine-Tuning (FT)** We implement a targeted fine-tuning approach where only the modified FFN layer in MeG is updated while freezing all other parameters. The optimization is SGD with a learning rate of 0.3. For GPT-J, we fine-tune the 9th FFN layer for 80 epochs with a batch size of 32. For Phi-2, we fine-tune the 29th FFN layer for 30 epochs with a batch size of 128. For Llama-3, we fine-tune the 14th FFN layer for 50 epochs with a batch size of 32.

**MALMEN** Our reproduction of MALMEN involves careful dataset re-partitioning to prevent data leakage as we changed the origin test set of ZsRE. Specifically, We remove all training examples where the 'src' field overlaps with test set questions in our ZsRE, creating a clean train-test split. Since COUNTERFACT lacks dedicated training data, we evaluate it using the ZsRE-trained hypernetwork. In our replication, we target the second linear layer of FFN modules across layers 26-31 in Phi-2, with epoch configurations of {1, 1, 10, 20} for edit scales of {1024, 2048, 4096, 10k} respectively. For Llama-3, the same set of FFN modules is targeted as Phi-2, with training schedules of {1, 1, 5, 10} epochs across the same four edit scales. For GPT-J, we maintain the original implementation's pattern with {1, 1, 5, 10} epochs at corresponding four edit scales. In 10k-scale experiments, the batch size is fixed at 25 to align with the structural requirements of the model, while all other hyperparameters remain consistent with the official implementation.

**SCEN** We reproduce SCEN with three key adjustments: selecting layer 29 for Phi-2 and layer 27 for GPT-J; using SGD as the optimizer instead of Adam; and assigning dataset-specific router thresholds (theta). We apply separate learning rates for index and expert networks: 0.005/0.08 for Phi-2 and 0.015/0.07 for GPT-J. On COUNTERFACT, theta is set to 0.65 for GPT-J and 0.75/0.70 for Phi-2 (1024/2048 edits). On ZsRE, theta is 0.85 for GPT-J and 0.75/0.70 for Phi-2, regardless of edit size. All other settings follow the official SCEN implementation.

Figure 8: Causal Tracing (using the method of ROME (Meng et al., 2022)). Each grid cell's intensity reflects the average causal indirect effect of a hidden state on the expression of a factual association, with strong causal mediators highlighted with darker colors. We observe that activations in layers 3 through 11 exhibit the most significant causal effects, indicating that these layers serve as key mediators in factual association expression.

**MEMIT and PMET** In the reproduction of MEMIT and PMET, we follow the default hyperparameter settings for the GPT-J model as specified in the original implementations. For the Phi-2 model, however, we apply the layer selection procedure provided by the MEMIT localization code. As shown in Fig. 8, this process identifies a set of sensitive layers. Based on this, we select layers 3, 4, 5, 6, 7, 8, 9, 10, 11 for MEMIT, and set the `mom2_update_weight` to 15000. For PMET, we use layers 3, 4, 5, 6, 7, 8, 9, 10, with `mom2_update_weight` set to 1000. Through empirical evaluation on sampled edit tasks, we observe that these layer configurations achieve the best overall performance across Reliability, Generality, and Locality, making them our final choice for experiments on Phi-2. For the Llama-3 model, we follow the same procedure. Specifically, for MEMIT we select layers 2, 3, 4, 5, and 6 with the mom2 update weight set to 1000, while for PMET we select layers 2, 3, 4, 5, and 6 with the mom2 update weight set to 13000.

### A.13 STATEMENT ON THE USE OF AI ASSISTANCE

In the preparation of this paper, we employed a Large Language Model (LLM) as a research and writing assistant. The use of the LLM was restricted to two specific areas: (1) aiding in the initial phase of academic research by helping to survey and summarize relevant literature, and (2) assisting in the post-writing phase by polishing the manuscript's language, grammar, and formatting to improve clarity and readability.

