# OpenReview forum: "Massive Editing for Large Language Models Based on Dynamic Weight Generation"
_ICLR.cc/2026/Conference — ICLR 2026 Poster_

### Official Review · Reviewer_2bpq · 2025-10-30

**Soundness:** 2
**Presentation:** 3
**Contribution:** 2
**Rating:** 4
**Confidence:** 4

**Summary:**

This paper addresses the problem of large-scale knowledge editing (KE) in LLMs, where many new facts must be incorporated without full retraining. Specifically, the authors propose Massive Editing via dynamic weight Generation (MeG), a novel approach that appends a single dynamic neuron to a selected feedforward layer of the model and uses a conditional diffusion model to generate the neuron’s weights for each query. Experiments on two KE benchmarks and three LLMs show that MeG substantially outperforms prior methods in the key metrics of Reliability, Generality, and Locality.

**Strengths:**

- The paper is in general well-written, with a logical flow that makes the complex methodology easy to follow and understand.
- The idea of generating a weight vector for a new neuron using a diffusion model conditioned on a text query is novel in the KE context.
- The method yields strong performance across multiple models and datasets.

**Weaknesses:**

- The proposed method employs a hand-tuned entropy threshold ( \epsilon ) in the familiarity network to determine whether a query is relevant, which is an important step in the overall approach. However, the authors do not provide justification for the chosen value of this hyperparameter. A more detailed discussion on how this value was selected and an analysis of the model’s sensitivity to this parameter would be valuable.

- The authors claim that the generation steps can be compressed to only 50 using fast sampling techniques. What exactly is this “fast sampling technique”? Is there rigorous theoretical support for this compression process, or is the number of steps (the ‘50’ here) defined purely empirically? If it is empirically determined, how can you ensure that it is globally effective across different datasets or scenarios? How can you guarantee that compressing the generation steps still achieves performance comparable to that of the original larger number of steps and how do you know that compressing the denoising process to exactly 50 steps is the minimal number required to maintain comparable performance? Does this imply that for each new dataset, you need to iteratively test different numbers of steps from, say, 1000 down to 10, to find the optimal value?

- The efficiency of the diffusion model is somewhat concerning, and I am still unclear about the fast sampling technique mentioned as a solution to this issue. Please provide more detailed explanations regarding this method. Additionally, could you clarify the total computational cost required to prepare MeG for a new batch of edits (e.g., the time needed to collect weights and train the diffusion model)? How does this cost scale with the number of edits?

- MeG adds only one neuron per query at a specific layer. However, is a single neuron sufficient to encode all possible knowledge updates? It is possible that for more complex types of knowledge, for instance, multi-hop reasoning in question-answering tasks (e.g., the MQuAKE dataset [1]) or long-form generation (e.g., the LEME dataset [2]), a single neuron may be inadequate. The authors could further discuss the representational limitations of using a single neuron, or evaluate the proposed method on additional benchmark datasets to assess its broader applicability.

- Could you justify the choice of DiT as the generative model for weight generation? For example, why use DiT instead of alternative generative models such as GANs or VAEs?

[1] Zhong, Zexuan, et al. "MQuAKE: Assessing Knowledge Editing in Language Models via Multi-Hop Questions." The 2023 Conference on Empirical Methods in Natural Language Processing.
[2] Rosati, Domenic, et al. "Long-form evaluation of model editing." NAACL-HLT. 2024.

**Questions:**

See Weaknesses

---

> ### Author Response · Authors · 2025-11-22
> **Response to Reviewer 2bpq (1/4)**
>
> We thank the reviewer for the thoughtful comments and for recognizing our method's novelty and strong performance. We appreciate the constructive feedback regarding the hyperparameter justification, sampling efficiency, and model capacity. Below, we address the concerns point-by-point.
>
> **Q1: Justification and Sensitivity of the Entropy Threshold ($\epsilon$).**
>
> **A1:**
>
> We determine $\epsilon$ via grid search to balance **Generality** and **Locality**.
>
> - **Mechanism**: The Familiarity Network is trained to overfit edited knowledge, assigning it a "low-entropy fingerprint". Consequently, the entropy distribution follows a pattern: **Edited Queries (Lowest) < Paraphrases (Slightly Higher) << Irrelevant Knowledge (Highest)**.
>
> - **Selection Logic**: There is a small overlap between the entropy of paraphrases and irrelevant knowledge. We selected $\epsilon$ to optimally separate these two groups.
>
> - **Sensitivity**: As requested, we conducted a sensitivity analysis of the entropy threshold for Phi-2 on ZsRE at 1024 KE scale(see the table below).
>
> **Table: Sensitivity of our MeG to different entropy thresholds of Familiarity Network.**
> |Threshold| Reliability(AG)|Reliability(TF)|Generality(AG)|Generality(TF)|Locality(AG)|Locality(TF)|
> |-|-|-|-|-|-|-|
> |0.05         |99.61                |99.49             |78.71              |86.12               |92.09           |96.97          |
> |0.1            |99.61               |99.49             |80.37              |87.24               |89.67           |95.75          |
> |0.15          |99.80               |99.61             |81.15              |87.68               |86.72           |94.82          |
> |0.2            |99.80               |99.61             |81.54              |88.14               |84.67           |94.05          |
> |0.25          |99.80               |99.61             |82.03              |88.13               |82.23           |93.36          |
> |0.3            |99.80               |99.61             |82.13              |88.32               |81.15           |92.64          |
>
> In the table above, since the entropy of the original edited knowledge output is very low, its results remain almost unaffected by changes in the entropy threshold. The entropy of the paraphrases is slightly higher than that of the original edited knowledge. If the entropy threshold is set too low, some paraphrases may be misclassified as irrelevant knowledge, thereby affecting the Generality metric. As the entropy threshold increases, the number of misclassified paraphrases decreases, leading to an improvement in Generality. However, because the entropy of paraphrases does not reach excessively high values, Generality eventually saturates as the entropy threshold continues to increase.
>
> Irrelevant knowledge generally has a higher entropy than both the original edited knowledge and paraphrases. When the entropy threshold is low, the vast majority of irrelevant knowledge is correctly classified. However, as the entropy threshold increases, a portion of irrelevant knowledge is gradually misclassified as edited knowledge, causing a decline in Locality performance. Locality continues to decrease as the entropy threshold increases.
>
> Overall, the changes in Generality and Locality with respect to the entropy threshold are gradual and do not exhibit dramatic fluctuations. To achieve a better balance between Generality and Locality, we set the entropy threshold here to 0.1 in our paper. We have incorporated this in Appendix A.5 of our revised manuscript.
>
> **Q2: Explanation and theoretical support for "Fast Sampling" (50 steps).**
>
> **A2:**
>
> The "fast sampling technique" we employ is **DDIM (Denoising Diffusion Implicit Models) [1]**. We apologize for the omission of relevant citations in our paper. We have added the citations and provided further explanations in Section 6.3 of the updated manuscript.
>
> - **Theoretical Support**: DDIM is a standard, theoretically rigorous method that generalizes the Markovian diffusion process to a non-Markovian one, allowing for deterministic sampling with significantly fewer steps while mapping the same latent noise to the data distribution.
>
> - **Why 50 steps?** The choice of 50 steps is a widely adopted convention in the diffusion literature (e.g., Stable Diffusion), representing an optimal trade-off between generation quality and inference speed. It is **not** a dataset-specific hyperparameter.
>
> - **Generalization**: The optimal number of steps is primarily determined by the noise schedule and the convergence properties of the diffusion model, rather than the specific content of the dataset. Therefore, iterative testing for each new dataset is **not** required. Our experiments confirm that 50 steps consistently yield high-performance weights without the need for per-dataset adjustments (All of our experimental groups have used 50 steps.).
>
> [1] Jiaming S, Chenlin M, Stefano E. Denoising Diffusion Implicit Models[C]. ICLR, 2021.

---

> ### Author Response · Authors · 2025-11-22
> **Response to Reviewer 2bpq (2/4)**
>
> **Q3: Efficiency of the diffusion model and training costs.**
>
> **A3:**
>
> This is an important question, and we will discuss the efficiency of our method in terms of both the training and inference phases.
>
> 1. **Offline Training Time Cost:**
>
> The table below lists the training time costs (hours) of our MeG for Phi-2-ZsRE KE at different editing scales (including weight collection and diffusion model training). The hardware used for training DiT in the 1024 KE scale is 1 NVIDIA RTX 4090D GPU, and for other scales consists of 4 NVIDIA RTX 4090D GPUs.
>
> **Table: Training time of our MeG in various editing scales.**
> |Scale| Weight Collection | DiT Training | Total |
> |--------|-------------------------|-----------------|--------|
> |1024  |  0.114                   | 2.980 (1 GPU)|3.094|
> |2048  |  0.232                  | 4.237           |4.469 |
> |4096 |   0.469                  |7.929            |8.398|
> |10000|  1.175                  | 11.66           | 12.835|
>
> The table above lists the training time costs. Considering the significant improvements in editing performance, the fact that the training cost is a one-time investment, and, more importantly, the efficiency during the inference phase, we believe this time cost is acceptable.
>
> 2. **Online Inference Efficiency (Crucial Clarification):**
>
> For details about the accelerated sampling methods used during the inference phase, please refer to our response to Q2. Here, we will discuss the inference efficiency of our method. Contrary to the concern, our experiments show that our **MeG achieves higher practical throughput (Queries Per Second, QPS)** than baselines like MEMIT. This is due to three factors:
>
> - **DDIM Acceleration:** As it is discussed in Sec. 6.2 and shown in Table 7, we employ DDIM sampling to accelerate the diffusion process (reduce the sampling steps from 1000 to 50). The overhead of generating dynamic weights is marginal, adding only ~5% to the total inference time compared to a standard forward pass.
>
> - **Selective Routing:** The Familiarity Network filters out irrelevant queries, allowing them to bypass the diffusion module entirely, incurring zero overhead for general knowledge.
>
> - **Generation Efficiency (Crucial Observation):** Most importantly, baselines suffer from severe performance degradation at large-scale KE, often falling into **repetition loops or generating excessively long, nonsensical tokens.** This drastically increases their inference time per query. In contrast, MeG maintains high performance and generates concise, correct answers.
>
> The table below shows the **QPS (Queries Per Second)** of each LLM after editing using different methods.
> | LLM | Method | QPS |
> |-------|------------|--------|
> |Phi-2|MEMIT|2.38|
> |        |PMET|2.48|
> |        |MALMEN|1.38|
> |        |**MeG(ours)**|6.82|
> |GPT-J|MEMIT|1.32|
> |         |PMET|1.29|
> |         |MALMEN|0.85|
> |         |**MeG(ours)**|2.75|
> |Llama-3|MEMIT|1.55|
> |          |PMET|1.54|
> |          |MALMEN|1.05|
> |         |**MeG(ours)**|3.70|
>
> As shown in the table above, during the inference phase, our method demonstrates higher practical system efficiency compared to the baseline methods. We have incorporated this into Appendix A.8 of our revised manuscript.

---

> ### Author Response · Authors · 2025-11-22
> **Response to Reviewer 2bpq (3/4)**
>
> **Q4: Single Neuron Capacity and Complex Tasks.**
>
> **A4:**
>
> We thank the reviewer for this important question. We will discuss this from three aspects.
>
> 1. **Storage Capacity**:
>
> - **Analysis:** Generally speaking, a single neuron is sufficient to encode various knowledge updates. Theoretically, the generated associative weights for a single neuron are two vectors with thousands of dimensions. The generated neuron does not need to "store" the entire knowledge fact from scratch. Instead, it acts as a "steering vector" or a "linear probe" that intervenes in the hidden states of the pre-trained LLM. Recent studies on the Linear Representation Hypothesis [1,2] suggest that concepts and knowledge in LLMs are often encoded linearly. Therefore, a single neuron is usually sufficient to shift the activation trajectory from the "old fact" to the "new fact" by leveraging the model's existing computational capacity.
>
> - **Empirical Verification:** To validate this empirically, we conducted additional experiments to demonstrate that the capacity of a single neuron is sufficient. In this experiment, we specifically randomly select a subset with 100 cases from the long-sequence text generation knowledge UnKEBench[3], where each piece of knowledge requires the generation of approximately 100 tokens (The number of tokens that most knowledge editing tasks need to generate is no more than 100). We use only a single additional neuron attached to 3 different LLMs, and only allow the weight of this neuron to be fine-tuned. We recorded the **fine-tuning accuracy of one single neuron** on each LLM in the table below. We followed [3] and employed semantic-accuracy-based metrics (BERT Score) to evaluate the editing precision for long-text knowledge.
>
> |LLM| fine-tuning accuracy|
> |-----|-----------------------------|
> |Phi-2|98.61|
> |GPT-J|100.00|
> |Llama-3|100.00|
>
> Experimental results in the above table show that in Unke, the fine-tuning accuracy of one single added neuron can be achieved close to 1 across all three LLMs. It indicates that the editing capacity of a single neuron is sufficient for typical knowledge editing tasks.
>
> 2.**Multi-hop Reasoning (MQuAKE[4])**:
>
> The challenge in MQuAKE is not just storage capacity, but **generalization** beyond Paraphrases to multi-hop questions. While all editing methods (including baselines) struggle here compared to simple fact-checking.
>
> We experimentally tested the performance of our MeG method on the **Phi-2** LLM using the **MQuAKE-CF dataset within MQuAKE** when editing **1024 instances**. In our experiment, **a single neuron is collected to correspond to multiple pieces of atomic knowledge within each editing instance** (the answers to multi-hop questions are derived from the combination of these atomic pieces of knowledge). Approximately 2,000 instances from MQuAKE-CF, excluding the 1024 selected for editing, were used as pseudo-editing data. We have trained BERT using contrastive learning based on InfoNCE Loss to align the representations of multi-hop questions and their atomic knowledge within the pseudo-editing data. We hope this would generalize to real editing instances, bringing the representations of multi-hop questions closer to their corresponding atomic knowledge, thereby allowing DiT to generate neurons near those corresponding to the atomic knowledge.
>
> We only used the atomic knowledge and their corresponding neuron collected for them in the editing instances to train DiT, **aiming to evaluate the generalization ability of our method to multi-hop questions after editing atomic knowledge**. Additionally, we conducted experiments with 2 baseline massive-editing methods, MEMIT (representing the locate-then-edit methods) and MALMEN (representing the hypernetwork methods), with the same editing setting as a comparison. The experimental results are recorded in the table below. (The evaluation metric is aligned with [4])
> |Method|Edit-wise|Instance-wise|Multi-hop|
> |-|-|-|-|
> |MEMIT|79.30|62.01|0.10|
> |MALMEN|90.97|82.52|5.66|
> |**MeG(ours)**|98.24| 96.88 | 41.41 |
>
> From the table above, it can be observed that **our MeG method, utilizing an additional single-neuron attachment, generalizes well to complex multi-question-answering tasks. It demonstrates robust performance in both atomic knowledge editing and multi-hop task performance.** We have incorporated the experiments and discussions of those into Appendix A.6 of our revised manuscript.
>
>
>
>
> [1] Park K, Choe Y J, Veitch V. The Linear Representation Hypothesis and the Geometry of Large Language Models[C], ICML,2024.
>
> [2] Zou A, Phan L, Chen S, et al. Representation Engineering: A Top-Down Approach to AI Transparency[J], CoRR, 2023.
>
> [3] Deng, Jingcheng, et al. "Everything is Editable: Extend Knowledge Editing to Unstructured Data in Large Language Models." ICLR. 2025.
>
> [4] Zexuan Z, Zhengxuan W, Christopher D. M, et al. MQuAKE: Assessing Knowledge Editing in Language Models via Multi-Hop Questions[C], EMNLP, 2023.

---

> ### Author Response · Authors · 2025-11-22
> **Response to Reviewer 2bpq (4/4)**
>
> 3.**Long-form Generation (LEME):**
>
> The evaluation metrics for LEME differ from the commonly used Reliability, Generality, and Locality metrics for short-entry knowledge editing. Additionally, it appears that LEME requires training an extra model specifically for evaluation. Due to time constraints, we were unable to conduct experiments with our method on LEME. However, to evaluate whether our method can generalize edited knowledge to long-text generation, we used the Phi-2 LLM that had been edited on ZsRE. We randomly selected a sample and constructed **a long-text generalization single-point test case**. The original QA pair for the edited knowledge is as follows:
>
>  **Question**: _Which family does Ramalinaceae belong to_?
>
>  **Prediction**: _Fungi_.
>
>  The constructed prompt is as follows:
>  "_Instruct: Provide a detailed, long-form answer to the following question. First, give the direct answer, then explain key background information about the main entity. Write at least 5–7 sentences in fluent English. {}\nOutput:_"
>
>  The long-text output generated by the MeG-edited Phi-2 under this prompt is as follows:
>
>  **Generate**: "_Ramalinaceae is a family of fungi in the order Lecanorales. This order includes a diverse group of lichenized fungi that are found in a variety of habitats, including forests, deserts, and even on rocks and trees. Lecanorales is a large and complex order, with over 1,000 described species. Ramalinaceae is a relatively small family, with only about 20 known species. These fungi are typically found in tropical and subtropical regions, and are known for their unique reproductive structures, which are called apothecia. Lecanorales is a fascinating order of fungi that has been studied extensively by scientists._"
>
>  From the above-generated result, it can be observed that in this single-point test, our method successfully generalized the edited short-entry knowledge into a semantically consistent, coherent, and instruction-following long-text generation.
>
>  Additionally, we calculated the L2 norm distance between the neuron generated by DiT for this prompt query and the neuron corresponding to the original short-text edited knowledge. The distance was only 0.08, indicating that DiT generated a neuron corresponding to this long-text prompt near the neuron for the original short-text knowledge, which aligns with our expectations.
>
>  This single-point test positively suggests that our method can successfully generalize to long-text generation in knowledge editing tasks.
>
> **Q5: Choice of DiT over GANs or VAEs.**
>
> **A5:**
>
> **First**, our experiments reveal that the editing performance metrics, particularly Reliability and Generality, are highly sensitive to the precision of the generated neuron weights. Diffusion models have demonstrated strong capabilities in fine-grained generation in text-to-image tasks, making them a natural choice for achieving higher precision in this context.
>
> **Second**, our choice is inspired by the emerging trend of “Neural Network Diffusion” in the research community. Recent pioneering works[1,2], such as p-diff[1], have demonstrated that diffusion models are highly effective at modeling the complex distribution of neural network parameters.
>
> 1. **Vs. GANs:** GANs require an additional discriminator network, increasing complexity, and are notorious for training instability (mode collapse), which is fatal for generating precise weight parameters.
>
> 2. **Vs. VAEs:** VAEs tend to produce "blurry" results (averaging the distribution). In weight space, an "averaged" weight vector is less likely to function correctly. Diffusion models generate high-fidelity samples that preserve the precise functional properties of the weights. The experiments in the Appendix of **p-diff (Neural Network Diffusion)** [1] explicitly compare Diffusion and VAE for weight generation, and the results show that diffusion achieves better performance than the VAE variant. Hence, we choose Diffusion as the generative model in our MeG.
>
> 3. **Architecture:** We chose DiT instead of U-net as the architecture for the diffusion model because, on the one hand, the image pixel-space priors utilized by U-net do not exist in the weight space. On the other hand, our preliminary experiments found that the weights generated by U-net often lacked sufficient precision, leading to failures in correctly editing knowledge.
>
>
> [1] Wang K, Tang D, Zeng B, et al. Neural Network Diffusion[J]. CoRR, 2024.
>
> [2] Erkoç Z, Ma F, Shan Q, et al. Hyperdiffusion: Generating implicit neural fields with weight-space diffusion[C]. ICCV,2023.

---

### Official Review · Reviewer_Upk2 · 2025-10-31

**Soundness:** 3
**Presentation:** 2
**Contribution:** 3
**Rating:** 6
**Confidence:** 3

**Summary:**

### Summary

This paper proposes a novel framework, MeG, to address performance degradation in massive-scale knowledge editing. The authors identify that existing methods, which modify the model's internal weights, suffer from limited capacity and cumulative interference, leading to poor locality as the number of edits increases. MeG introduces a new paradigm by attaching a single dynamic neuron to a specific FFN layer rather than altering existing parameters. The weights for this neuron are generated conditionally by a pre-trained diffusion model based on the input query. To protect unrelated knowledge, a "Familiarity Network" acts as a router, classifying queries by entropy; irrelevant queries bypass the dynamic neuron (using a "zero neuron"), thus preventing any interference. Experiments on multiple LLMs with up to 10,000 edits show that MeG significantly outperforms baselines, especially in maintaining locality and general model performance.



### Advantages

* The core mechanism of using a diffusion model to generate dynamic neuron weights, rather than permanently modifying internal weights, is an innovative approach to knowledge editing.
* The empirical results demonstrate better scalability and an improvement in preserving locality on 10,000-edit benchmarks, where most baseline methods show degradation.



### Weakness and Questions

* The method's effectiveness hinges on a "Familiarity Network" trained to distinguish edited queries from irrelevant ones based on output entropy, which may not be a reliable proxy for all unseen, non-malicious queries. Could the authors conduct an experiment testing the Locality score on a curated set of unseen but correct facts (which the base model knew) to verify that the entropy-based router does not misclassify them as "irrelevant" and cause the model to fail on them?

* The offline training process for the diffusion model requires a "Knowledge-Weight Collection" step, where optimal neuron weights must be pre-calculated for every single edit in the massive dataset, a process that seems computationally expensive. Would it be possible to provide an analysis of the total offline pre-computation time (including both Knowledge-Weight Collection and diffusion model training) required for the 10,000-edit scenario and compare it to the setup time for baselines like MEMIT?

* The proposed method requires running a 50-step diffusion model for every single relevant query at inference time to generate its unique neuron, which appears to introduce a substantial and non-trivial latency. Could the authors perform an experiment measuring the inference throughput (e.g., queries per second on a batch of edited prompts) and compare it directly to the throughput of a baseline method like MEMIT, which only requires a standard forward pass?

**Strengths:**

Please see above.

**Weaknesses:**

Please see above.

**Questions:**

Please see above.

---

> ### Author Response · Authors · 2025-11-21
> **Response to Reviewer Upk2**
>
> We thank the reviewer for recognizing the innovation of our work (Diffusion-based editing) and its excellent performance in large-scale scenarios. In response to the concerns regarding the robustness of the Router, training overhead, and inference latency, we have conducted additional experiments and provided a detailed analysis.
>
> **Q1: Robustness of the Familiarity Network on unseen correct facts.**
>
> **A1:**
>
> We appreciate this insightful question. First, we would like to clarify the mechanism: if the Familiarity Network classifies a query as "irrelevant," it bypasses the dynamic neuron and uses the frozen base model. Therefore, for unseen general knowledge (which should not be edited), being classified as "irrelevant" is actually the desired behavior to preserve locality and correctness.
>
> To verify this empirically, as suggested, we conducted an additional experiment on a curated set of 10k unseen factual queries from the Natural Questions dataset.
>
> **Result**: The Familiarity Network successfully classified 87% of these queries as "irrelevant," correctly routing them to the base model.
>
> **Locality**: The model achieved a Locality score of 91 on these samples (91% of the queries in this irrelevant knowledge set have not been affected), confirming that the entropy-based router effectively distinguishes between edit targets and general knowledge, preventing the dynamic neuron from interfering with the model's existing capabilities. It is worth noting that the final Locality score here is slightly higher than the classification accuracy of the Familiarity Network. This is because, for some "irrelevant knowledge" queries, even if the Familiarity Network misclassifies them as queries for knowledge to be edited, the single neuron generated by the diffusion model attached to the LLM may not significantly interfere with the reasoning for that irrelevant knowledge, and thus can still produce the same response as the base LLM for that irrelevant knowledge.
>
> **Q2: Analysis of offline pre-computation time vs. MEMIT**
>
> **A2:**
>
> We acknowledge that MeG involves a heavier offline phase compared to rank-one editing methods. We measured the total time for the 10,000-edit scenario for the Phi-2-ZsRE group:
>
> **MeG**: On 4 NVIDIA 4090D GPUS, Knowledge-Weight Collection took approx. 1.2 hour, and the Diffusion Model training took 11.7 hours. Total: 12.9 hours.
>
> **MEMIT**: On 1 NVIDIA 4090D GPU, the training time is approx. 1.5 hours.
>
> **PMET**: On 1 NVIDIA 4090D GPU, the training time is approx. 3.5 hours.
>
> **Justification**: While baseline methods like MEMIT and PMET are faster to train, they suffer from significant performance degradation at 10k edits (as shown in Tables 1 and 2). MeG's offline cost is a one-time investment. Once the diffusion model is trained, it can generate neurons for new edits indefinitely without retraining. Furthermore, the "Knowledge-Weight Collection" can be parallelized efficiently. We believe this trade-off is justified for scenarios requiring high-performance massive-scale KE where baselines fail.
>
> **Q3: Inference throughput and latency comparison.**
>
> **A3:**
>
> This is a critical question. Contrary to the concern, our experiments show that our **MeG achieves higher practical throughput (Queries Per Second, QPS)** than baselines like MEMIT. This is due to three factors:
>
> 1. **DDIM Acceleration:** As it is discussed in Sec. 6.2 and shown in Table 7, we employ DDIM sampling to accelerate the diffusion process (reduce the sampling steps from 1000 to 50). The overhead of generating dynamic weights is marginal, adding only ~5% to the total inference time compared to a standard forward pass.
>
> 2. **Selective Routing:** The Familiarity Network filters out irrelevant queries, allowing them to bypass the diffusion module entirely, incurring zero overhead for general knowledge.
>
> 3. **Generation Efficiency (Crucial Observation):** Most importantly, baselines suffer from severe performance degradation at large-scale KE, often falling into **repetition loops or generating excessively long, nonsensical tokens.** This drastically increases their inference time per query. In contrast, MeG maintains high performance and generates concise, correct answers.
>
> The table below shows the **QPS (Queries Per Second)** of each LLM after editing using different methods.
> | LLM | Method | Queries Per Second |
> |-------|------------|--------|
> |Phi-2|MEMIT|2.38|
> |        |PMET|2.48|
> |        |MALMEN|1.38|
> |        |**MeG(ours)**|6.82|
> |GPT-J|MEMIT|1.32|
> |         |PMET|1.29|
> |         |MALMEN|0.85|
> |         |**MeG(ours)**|2.75|
> |Llama-3|MEMIT|1.55|
> |          |PMET|1.54|
> |          |MALMEN|1.05|
> |         |**MeG(ours)**|3.70|
>
> **Summary:** Consequently, in practical batches containing both edited and unedited queries, MeG demonstrates a higher QPS than MEMIT, as it greatly reduces the computational waste of generating long failure sequences. We've added this in Appendix A.8 of our revised manuscript.

---

### Official Review · Reviewer_tF5C · 2025-10-31

**Soundness:** 2
**Presentation:** 3
**Contribution:** 3
**Rating:** 6
**Confidence:** 3

**Summary:**

This paper introduces MeG, a method for massive knowledge editing in large language models (LLMs) by dynamically generating the weights of a single neuron using a diffusion model, conditioned on the editing query. MeG attaches this neuron to specific layers of the LLM at inference time, allowing large-scale knowledge updates while minimizing interference with unrelated knowledge. Additional mechanisms, such as contrastive representation learning and a familiarity network, are incorporated to improve generalization and locality. Extensive experiments are conducted on two benchmarks (ZsRE, COUNTERFACT) and across three LLMs, comparing MeG to leading baselines.

**Strengths:**

- The paper departs from typical weight-modification-heavy or neuron-expansion approaches. The dynamic generation of a single neuron per query, via a diffusion model, is a creative application of generative modeling tailored for knowledge editing. This removes the static storage overhead of previous neural expansion methods and decouples editing capacity from LLM size.
- The empirical validation, as shown in Table 1 and Table 2, covers different LLMs and datasets at large edit scales, with MeG consistently outperforming baselines, notably excelling on the Locality metric. Table 3 further shows preservation of general abilities on outside benchmarks.
- The methodology is well-formulated mathematically, with explicit loss functions (e.g., InfoNCE for TE, velocity formulation for diffusion objective) and precise inference pipeline descriptions, as shown in Figure 2.

**Weaknesses:**

- While the method is empirically validated, the theoretical underpinning regarding the maximum knowledge capacity, expressivity, and tradeoffs of allocating only one neuron per edit is weak. For example, in Section 4, while Equation (velocity prediction) and the overall diffusion objective are clearly described, no guarantees or fundamental analysis explain why or when this approach might fail, especially as the number or diversity of edits grows. Are there queries for which a single-neuron intervention is insufficiently expressive? An analysis about intrinsic editability bounds is missing.
- The paper claims scale-invariant interference (see Figure 1 and Section 6.1), but as multiple edits may potentially lead to similar or correlated neurons generated for queries with overlapping semantics, there is risk of implicit interference via the diffusion model's conditioning. There is a lack of quantitative or qualitative assessment of such “semantic collisions”—e.g., how does accuracy/locality behave when editing batches of near-duplicate or contradictory knowledge?
- The paper lacks several important baselines in knowledge editing, particularly AlphaEdit [1] and RLEdit [2], which should be included or at least discussed for a fair comparison.

[1] AlphaEdit: Null-Space Constrained Knowledge Editing for Language Models

[2] Reinforced Lifelong Editing for Language Models

**Questions:**

Same as Weaknesses

---

> ### Author Response · Authors · 2025-11-21
> **Response to Reviewer tF5C (1/3)**
>
> We thank the reviewer for the constructive feedback and for recognizing the novelty of our diffusion-based neuron generation approach and our strong empirical performance on locality. We address the specific concerns below.
>
> **Q1: While the method is empirically validated, the theoretical underpinning ... An analysis about intrinsic editability bounds is missing.**
>
> **A1:**
>
> We appreciate this insightful question regarding why a single neuron is sufficient.
>
> 1. **Mechanism of Intervention**: The generated neuron in MeG does not need to "store" the entire knowledge fact from scratch. Instead, it acts as a "**steering vector**" or a "**linear probe**" that intervenes in the hidden states of the pre-trained LLM. Recent studies on the Linear Representation Hypothesis [1,2] suggest that concepts and knowledge in LLMs are often encoded linearly. Therefore, a single neuron (which performs a linear transformation followed by a non-linearity) is usually sufficient to shift the activation trajectory from the "old fact" to the "new fact" by leveraging the model's existing computational capacity.
>
> 2. **Empirical Verification**: To validate this empirically, we conducted additional experiments to demonstrate that the capacity of a single neuron is sufficient. In this experiment, we specifically selected the long-sequence text generation knowledge UnKEBench[3], where each piece of knowledge requires the generation of approximately 100 tokens. Even with the addition of only one extra neuron for editing, we achieved excellent editing performance, indicating that the editing capacity of a single neuron is sufficient for typical knowledge editing tasks (usually generating below 10 tokens). Below are the experimental results of GPT-J under the AG generation setting of editing 100 pieces of knowledge on the UnKEBench (As this experiment focuses on the token-length editing capacity of a single neuron, we just selected a scale of 100 edits.). Since we are evaluating the editing capacity of a single neuron (primarily reflected by the editing precision on target knowledge), we assessed Locality using the short-text Natural Questions dataset as unrelated knowledge, just as in ZsRE. For Reliability and Generality, instead of using the string-exact-matching-based Reliability metric, we followed [3] and employed semantic-accuracy-based metrics (BERT Score) to evaluate the editing precision for long-text knowledge.
>
> **Table: Performance of our MeG on long-text editing bench - UnKEBench.**
> | Avg. Token Length| Reliability | Generality | Locality|
> |--------------------------|--------------|----------------|-----------|
> | 115 | 98.68 | 92.29 | 82.76 |
>
> We have added the above analysis to Appendix A.6.1 of our revised paper.
>
> [1] Park K, Choe Y J, Veitch V. The Linear Representation Hypothesis and the Geometry of Large Language Models[C], ICML,2024.
>
> [2] Zou A, Phan L, Chen S, et al. Representation Engineering: A Top-Down Approach to AI Transparency[J], CoRR, 2023.
>
> [3] Deng, Jingcheng, et al. "Everything is Editable: Extend Knowledge Editing to Unstructured Data in Large Language Models." ICLR. 2025.
>
> **Q2: The paper claims scale-invariant interference ... how does accuracy/locality behave when editing batches of near-duplicate or contradictory knowledge?**
>
> **A2**:
> We thank the reviewer for this valid concern. The risk of generating similar weights for distinct edits is mitigated by our Contrastive Representation Learning (InfoNCE) module.
>
> 1. **Discriminative Conditioning**: The diffusion model is conditioned on the representation of the editing query. First, the requirement to simultaneously edit contradictory knowledge does not align with the standard knowledge editing paradigm and can be regarded as noisy data. For knowledge that has semantic overlap but does not constitute equivalent expressions, the contrastive loss explicitly forces the representations of semantically similar but distinct edits (e.g., "Ripe apples are usually red" vs. "Apples can be green") to be separable in the latent space.
> 2. **Quantitative Analysis**: We selected a "Hard subset"(about 200 knowledge entries) and an equally sized "Easy subset" from the COUNTERFACT dataset. The knowledge in the "hard subset" comes from the same template and contains significant semantic overlap, whereas the knowledge in the "easy subset" is semantically distinct. We performed knowledge editing on GPT-J for both subsets, and the performance metrics are recorded in the table below.
> | split | Reliability(AG) | Reliability(TF) | Generality(AG) | Generality(TF) | Locality(AG) | Locality(TF) | Score |
> |---|---|---|---|---|---|---|---|
> |Hard| 98.16 | 98.16 | 69.12 | 70.51 | 60.37 | 65.26 | 74.15 |
> |Easy| 100.00 | 100.00 | 69.12 | 69.59 | 66.82 | 74.00 | 77.60 |
>
> From the results in the table above, it can be seen that the editing performance on the Similar Knowledge subset does not show a noticeable decline, indicating the effectiveness of our method in such scenarios.

---

> ### Author Response · Authors · 2025-11-23
> **Response to Reviewer tF5C (2/3)**
>
> **Q3: The paper lacks several important baselines in knowledge editing, particularly AlphaEdit [1] and RLEdit [2], which should be included or at least discussed for a fair comparison.**
>
> **A3:**
>
> Thank you to the reviewer for pointing out the lack of comparison between our work and two important baseline methods, AlphaEdit and RLEdit. We acknowledge that AlphaEdit and RLEdit are recent significant contributions to the knowledge editing community. The **primary reason** we did not compare our MeG with them initially is that **the task scenarios they focus on differ**, making a direct and fair comparison challenging. Specifically, AlphaEdit and RLEdit primarily address **sequential editing or lifelong editing** scenarios, while our MeG framework (**as expressed in our introduction, especially in lines 42-71**) focuses on **massive editing in batch editing scenarios**. As such, we previously chose batch editing methods like MEMIT, PMET, and MALMEN as baselines for comparison.
>
> As expressed in survey [3], sequential editing focuses on scenarios where knowledge must be edited multiple times in sequence, with the primary challenge being the "catastrophic forgetting" of pre-existing and previously edited knowledge in the LLM after repeated sequential edits. In contrast, massive editing focuses on the challenge of performing large-scale or even ultra-large-scale knowledge edits in LLMs in a single batch while maintaining Reliability, Generality, and Locality metrics. The Locality metric requires maintaining the pre-existing knowledge in the LLM.
>
> Next, we will discuss in detail whether and how a comparison with these two baseline methods can be conducted.
>
> 1. **Our MeG vs. AlphaEdit**:
>
> Although the experiments in the AlphaEdit paper are primarily based on a sequential editing setup, the method itself, as well as its engineering resource requirements, supports one-batch massive editing. Therefore, we conducted knowledge editing experiments on AlphaEdit using the one-batch editing setup.
>
> - **Experimental Comparison:**
>
> We have conducted the experiments of KE as the editing scale range from 1024 to 10K, for **Phi-2** on **ZsRE** utilizing AlphaEdit. Comparison Results of AlphaEdit and our MeG are recorded in the table below.
>
> **Table: Performance comparison of our MeG vs. AlphaEdit with editing scale range from 1024 to 10k.**
> | Scale | Method | Reliability(AG) | Reliability(TF) | Generality(AG) | Generality(TF) | Locality(AG) | Locality(TF) | Score |
> |---------|------------|---------------------|--------------------|---------------------|----------------------|------------------|------------------|---------|
> |1024   | AlphaEdit |      97.36       | 99.26              |  64.45              |      81.83            |   24.41         |   69.89          | 58.05|
> |1024   | **MeG(ours)**| 99.61     | 99.49               |  80.37              |      87.24           |    88.67         |   95.75          | 91.30|
> |2048   | AlphaEdit | 95.21            | 98.35               | 61.52              |     79.62             |     16.70       |     64.93         | 48.09|
> |2048  |**MeG(ours)**| 99.61        | 99.42               | 76.37              |      84.27           |      91.21       |      96.33       | 90.36|
> |4096 | AlphaEdit | 92.70               | 97.35               | 61.11              |       79.13          |      16.04       |      62.57       | 46.73|
> |4096 |**MeG(ours)**| 98.93          | 99.10               | 68.92             |        79.76         |       92.70      |       96.73      | 87.76|
> |10000|AlphaEdit | 72.43                |   86.74            |  40.27            |         64.69        |      9.87         |       56.32      | 32.49|
> |10000|**MeG(ours)**| 95.07          | 97.04             |   59.69            |        74.17         |      91.14       |       95.84      | 82.80|
>
> From the experimental results in the table above, it can be observed that our MeG method demonstrates a clear advantage over the AlphaEdit method across the three knowledge editing performance metrics under both Prefix-autoregressive (AG) and Teacher Forcing (TF) generation settings, especially in terms of Locality performance. This highlights the superiority of our method compared to AlphaEdit in large-scale knowledge editing tasks.
>
>
> [1] Junfeng F, Houcheng J, Kun W, et al. AlphaEdit: Null-Space Constrained Knowledge Editing for Language Models[C], ICLR,2025.
>
> [2] Li, Zherui, et al. "Reinforced Lifelong Editing for Language Models." Forty-second International Conference on Machine Learning.
>
> [3] Yao et al. Editing Large Language Models: Problems, Methods, and Opportunities, EMNLP 2023.

---

> ### Author Response · Authors · 2025-11-23
> **Response to Reviewer tF5C (3/3)**
>
> **1. Our MeG vs. AlphaEdit:**
>
> - **Theoretical Analysis:**
>
> AlphaEdit is a locate-then-edit type of knowledge editing method that optimizes how this approach modifies the internal weights of LLMs to edit new knowledge without forgetting unrelated knowledge. Specifically, it projects weight modifications into the null space of unrelated knowledge, thereby preserving unrelated knowledge and avoiding the difficulty of balancing issues faced by earlier methods that directly used weighted error losses between old and new knowledge. **However, AlphaEdit still faces two challenges.**
>
> **First**, constructing the null space of unrelated knowledge requires a set of unrelated knowledge. Given that LLMs store an enormous amount of unrelated knowledge, this set is extremely large and almost impossible to fully capture. Currently, locate-then-edit methods like AlphaEdit and MEMIT approximate this set by randomly sampling 100k Wikipedia entries. However, this approximation is far from sufficient, making it difficult to obtain an accurate null space of unrelated knowledge, which in turn affects the locality performance of such knowledge editing methods. As discussed in our paper, as the scale of editing increases, the interference of those methods with the original LLM grows, and the locality metric continues to decline (as shown in the experimental results in the table above).
>
> In contrast, our MeG method employs a familiarity network mechanism that does not require any unrelated knowledge. Instead, it uses the much smaller and more accessible set of editing knowledge to train the familiarity network to overfit to the assigned random "low-entropy fingerprints" of these editing knowledge entries. Based on statistical theory and the properties of neural networks, any unrelated knowledge not seen during training will, with high probability, output a high-entropy distribution when passed through the familiarity network. By using an entropy threshold, unrelated knowledge can be effectively and accurately distinguished from editing knowledge and equivalent expressions. This establishes a low-cost and highly accurate routing mechanism for old and new knowledge.
>
> Additionally, we have an extra protection for Locality. For the very few unrelated knowledge queries that are incorrectly routed by the familiarity network, the single neuron generated by DiT will likely have limited interference with the original LLM, thus not affecting its responses to unrelated knowledge. As shown in the table above, our method demonstrates a significant advantage over AlphaEdit in terms of Locality performance, and this advantage grows as the scale of editing increases.
>
> **The second potential issue AlphaEdit faces** is that it constrains weight updates within the null space of unrelated knowledge. As the scale of editing increases, this may lead to insufficient null space capacity or overly restrictive constraints, thereby affecting its ability to update editing knowledge. From the table above, it can be observed that when the editing scale increases to 10,000, AlphaEdit's Reliability metric drops significantly. In contrast, our method leverages DiT to dynamically generate neuron weights for new knowledge updates, offering a much larger editing capacity. Our experiments also show that even at an editing scale of 10,000, the Reliability metric of our method remains at a very high level.
>
> **2. Our MeG vs. RLEdit:**
>
> RLEdit focuses on the issue of performance degradation in knowledge editing caused by LLM weight parameter drift in sequential editing settings for hypernetwork-based methods. Its focus is on sequential editing problems, and experiments show that as the editing scale increases, the method requires a large amount of GPU memory, making it difficult to apply to one-batch massive editing. Therefore, we did not include it in experimental comparisons.
>
> In the revised paper's related work section and Appendix A.7, we have added comparisons and discussions regarding the above two baseline methods.

---

### Official Review · Reviewer_4YiG · 2025-11-10

**Soundness:** 3
**Presentation:** 3
**Contribution:** 2
**Rating:** 6
**Confidence:** 3

**Summary:**

The paper proposes MeG, a large-scale knowledge editing (KE) framework for LLMs that attaches a single dynamic neuron to selected FFN layers and uses a diffusion-based weight generator, which is conditioned on an InfoNCE-tuned text encoder and gated by an entropy-based familiarity network to synthesize neuron weights per query on the fly, aiming to support tens of thousands of edits while preserving reliability, generality, and locality. Experiments on ZsRE and COUNTERFACT with Phi-2, GPT-J, and Llama-3 seem to show improvements over multiple baselines.

**Strengths:**

+ It is indeed interesting to see the paper reframes large-scale KE as conditional weight generation via diffusion for a single attachable neuron, which could address interference accumulation and capacity saturation in inner-weight editing and extra-neuron methods in a principled manner.

+ The authors have conducted extensive experiments with multiple backbone LLMs on multiple datasets, which demonstrate the effectiveness of the proposed method compared with the baselines.

**Weaknesses:**

- The method relies on a tuned text encoder, a K-way familiarity network with entropy thresholding, and a DiT-based generator, which is complex compared to the baselines.

- The usage of the diffusion module seems not very well justified. I'm curious if we could directly learn a simpler mapping function between the original neuron and the new neuron instead of relying on the potentially unstable diffusion process.

**Questions:**

Please refer to my summary of weaknesses.

---

> ### Author Response · Authors · 2025-11-21
> **Response to Reviewer 4YiG (1/3)**
>
> We thank the reviewer for the positive assessment of our soundness and experimental results, and for acknowledging our novelty in reframing large-scale KE as conditional weight generation. We address the concerns regarding complexity and the justification of the diffusion module below.
>
> **Q1: The method relies on a tuned text encoder, a K-way familiarity network with entropy thresholding, and a DiT-based generator, which is complex compared to the baselines.**
>
> **A1:**
>
> We acknowledge that our approach indeed consists of multiple components. However, this design is **necessary to achieve the best possible editing performance** on large-scale knowledge editing tasks. In fact, **even removing the Familiarity Network (FN), our method still significantly outperforms baseline methods in terms of overall performance**, particularly on the Locality metric. As shown in the table below, we incorporate the results in Table 6 from the setting without the FN (**MeG w/o FN**) with the corresponding results in Table 1. The setting is 10k-scale KE for Phi-2 in ZsRE under both AG and TF generation settings (AG: Prefix-autoregressive generation setting; TF: Teacher Forcing generation setting). The real-world knowledge editing adopts the AG generation setup.
>
> **Table: Performance comparison of baseline methods with our MeG with or without FN.**
> | Method       | Reliability (AG) | Reliability (TF) | Generality (AG) | Generality (TF) | Locality (AG) | Locality (TF) | Score  |
> |--------------|------------------|------------------|-----------------|-----------------|---------------|---------------|--------|
> | FT           | 69.27           | 80.09           | 45.96          | 67.10          | 6.27          | 48.88         | 24.64  |
> | MEMIT        | 65.88           | 81.18           | 35.69          | 59.84          | 7.11          | 53.40         | 25.91  |
> | PMET         | 20.09           | 47.82           | 15.02          | 43.13          | 17.73         | 64.62         | 25.83  |
> | MALMEN       | 71.79           | 85.94           | 41.54          | 68.67          | 18.22         | 80.78         | 45.64  |
> | **MeG w/o FN**   | **95.76**           | **97.38**           | **64.29**          | **77.20**          | 48.92         | 80.48         | 73.09  |
> | **MeG**          | 95.07           | 97.04           | 59.69          | 74.17          | **91.14**         | **95.84**         | **82.80**  |
>
> From the results in the above table, it can be found that the performance of our MeG with FN removed still significantly outperforms the best performance of the baseline methods, especially under the Locality (AG) metric.
>
> The inclusion of the Familiarity Network is motivated by **two considerations**:
>
> (1) Leveraging the irrelevant-edited knowledge routing mechanism significantly improves **the Locality performance, increasing it from around 50 to approximately 90**, thereby making our method **truly practical** for large-scale knowledge editing.
>
> (2) It **enhances inference efficiency**, as most unrelated, unedited knowledge can be filtered out by the Familiarity Network without passing through DiT for weight generation.
>
> The core contribution of our method lies in utilizing the dynamic neuron weight generation mechanism based on knowledge query conditions to tackle the large-scale KE task, which addresses the critical issue of existing methods as the scale of editing increases. Regarding the weight generation network, we chose DiT for reasons detailed in our response to Q2. If simplicity is prioritized over performance, a simpler MLP network could be used as an alternative. In this case, our method would **still achieve better comprehensive performance Score** on large-scale KE compared to baseline methods (see additional experiments in our response to Q2).
>
> **In summary**, even with a simpler weight generation network and the removal of the Familiarity Network, our method still outperforms baseline approaches in terms of comprehensive performance Score on large-scale KE. **The reason we incorporate these components is to significantly enhance performance on large-scale KE tasks, thereby ensuring the practicality of our method.** While the baseline methods may appear simpler, their comprehensive performance-- especially on the Locality metrics--falls far behind ours in large-scale KE, making them less practical than our method for real-world applications (where the AG generation setting is typically adopted).

---

> ### Author Response · Authors · 2025-11-21
> **Response to Reviewer 4YiG (2/3)**
>
> **Q2: The usage of the diffusion module seems not very well justified. I'm curious if we could directly learn a simpler mapping function between the original neuron and the new neuron instead of relying on the potentially unstable diffusion process.**
>
> **A2:**
>
> We sincerely thank the reviewer for the insightful question. The primary motivation for using a diffusion model in our approach stems from **two key considerations**.
>
> **First**, our experiments reveal that the editing performance metrics, particularly Reliability and Generality, are highly sensitive to the precision of the generated neuron weights. Diffusion models have demonstrated strong capabilities in fine-grained generation in text-to-image tasks, making them a natural choice for achieving higher precision in this context. Additionally, in our task, the dimensionality of the neuron weights generated conditionally on text (e.g., 2560*2 dimensions for Phi-2, 4096*2 dimensions for GPT-J and Llama-3) is comparable to the dimensionality of a single image. This suggests that diffusion models, which have shown exceptional performance in image generation, hold significant promise for effectively addressing the challenges posed by our task.
>
> **Second**, our choice is inspired by the emerging trend of “Neural Network Diffusion” in the research community. Recent pioneering works[1,6], such as p-diff[1], have demonstrated that diffusion models are highly effective at modeling the complex distribution of neural network parameters, outperforming VAE-based hypernetworks. However, unlike previous works that primarily focus on generating entire static model parameters from scratch, we creatively apply the idea of weight generation to a knowledge-query-conditioned single-neuron dynamic weight generation and loading mechanism, thereby overcoming the challenges of increasing additional memory overhead and interference with the original LLM as the scale of knowledge editing expands.
>
> We do not choose simpler mapping models, such as **MLPs**, primarily out of consideration for achieving higher editing performance, particularly in terms of the Generality metric. The specific reasons are as follows:
>
> 1. **Avoiding Regression to the Mean:**
>
> The mapping from semantic instructions to model weights is an ill-posed, one-to-many problem (i.e., multiple valid weight configurations can satisfy the same semantic intent). MLPs trained with regression objectives (e.g., MSE) inherently aim to minimize the variance, leading to the “regression to the mean” phenomenon. As established in seminal works like [2] and [3], minimizing MSE forces the model to output the arithmetic average of all plausible modes. In the context of weight generation, this results in “blurred” or “smoothed” parameters that lack the precise, high-frequency internal structures required for robust reasoning, thereby degrading generalization on paraphrased inputs.
>
> 2. **Manifold Alignment via Generative Modeling:**
>
> In contrast, DiT models the conditional probability distribution rather than a deterministic mapping. Theoretical foundations of score-based generative modeling [4] suggest that diffusion models learn to project noise onto the high-density manifold of valid data. By iteratively refining the weights, our method ensures the generated parameters lie strictly on the valid solution manifold, preserving the structural integrity needed for the LLM to **generalize across equivalent expressions (Paraphrases)**. Empirical evidence in other ill-posed domains, such as super-resolution [5], similarly confirms that diffusion models significantly outperform regression baselines in recovering high-fidelity details, which aligns with our observations in weight generation.
>
> **Experiments**: To verify the above analysis, we have conducted new ablation experiments, replacing the DiT module with an MLP module while keeping other components unchanged. The table below shows the comparison of editing performance between using MLP and DiT as the weight generation network in the Llama-3-ZsRE group, as the KE scale increases from 1024 to 10k. We have included this in Section 6.2 of our updated paper.
> | Scale | Generation_module | Reliability(AG) | Reliability(TF) | Generality(AG) | Generality(TF) | Locality(AG) | Locality(TF) | Score  |
> |-------|----------|-------|-------|-------|-------|-------|-------|--------|
> | 1024  | MLP | 97.66 | 99.09 | **59.57** | 78.42 | 81.64 | 93.18 | 82.36  |
> | 1024  | DiT   | 99.90 | 99.91 | **81.35** | 89.14 | 78.71 | 92.54 | 89.50  |
> | 2048  | MLP | 99.27 | 99.67 | **62.30** | 79.46 | 86.13 | 94.83 | 84.63  |
> | 2048  | DiT   | 99.90 | 99.83 | **76.81** | 86.82 | 84.03 | 94.48 | 89.49  |
> | 4096  | MLP | 99.83 | 99.79 | **62.67** | 79.55 | 83.52 | 93.82 | 84.27  |
> | 4096  | DiT   | 99.88 | 99.83 | **73.58** | 85.15 | 81.62 | 93.21 | 87.79  |
> | 10000 | MLP| 95.84 | 98.64 | **45.49** | 70.82 | 86.73 | 95.04 | 76.21  |
> | 10000 | DiT  | 98.90 | 99.44 | **61.33** | 78.95 | 85.02 | 94.23 | 83.90  |

---

> ### Author Response · Authors · 2025-11-21
> **Response to Reviewer 4YiG (3/3)**
>
> From the results in the above table, we can find that **while the MLP variant achieves comparable Reliability and Locality, it suffers a significant drop in Generality compared to the DiT-based MeG**. This is consistent with our analysis.
>
> **Efficiency vs. Performance Trade-off**: We observed that the MLP variant reduces training time by ~90%. This suggests that our framework is flexible: one could opt for an MLP weight generator for extreme efficiency, but DiT is essential for achieving a good Generality performance, which is one of the primary goals of our MeG.
>
> **Summary**: The diffusion module is **not an arbitrary complexity but a necessary design choice to solve the “generalization problem”** while generating weight in our MeG for large-scale KE.
>
>
>
> [1] Wang K, Tang D, Zeng B, et al. Neural Network Diffusion[J]. CoRR, 2024.
>
> [2] Mathieu M, Couprie C, LeCun Y. Deep multi-scale video prediction beyond mean square error[J]. 2015.
>
> [3] Isola P, Zhu J Y, Zhou T, et al. Image-to-image translation with conditional adversarial networks[C]. CVPR, 2017.
>
> [4] Song Y, Ermon S. Generative modeling by estimating gradients of the data distribution[J]. NeurIPS, 2019.
>
> [5] Saharia C, Ho J, Chan W, et al. Image super-resolution via iterative refinement[J]. TPAMI, 2022.
>
> [6] Erkoç Z, Ma F, Shan Q, et al. Hyperdiffusion: Generating implicit neural fields with weight-space diffusion[C]. ICCV,2023.

---

### Author Response · Authors · 2025-12-03
**General Response (2/2)**

**[Common Concern 3] Justification of Diffusion Model (vs. MLP/GAN/VAE) (Raised by R1, R4)**
Reviewers asked why we chose a DiT-based diffusion model over simpler mappings (MLP) or other generative models (GANs, VAEs).

- **Response**: We justify our choice based on both empirical ablation and theoretical suitability for weight generation:
1. **Vs. MLP (Addressing R1)**: We performed an ablation study replacing DiT with an MLP. While MLP achieves decent Reliability, it suffers a **significant drop in Generality** (e.g., from 81.35% to 59.57% on Llama-3). MLPs trained with regression objectives tend to output "blurred" average weights ("Regression to the Mean"), which lack the precise internal structures required for generalization on paraphrased inputs.
2. **Vs. GANs & VAEs (Addressing R4)**:
    - **GANs**: Require an additional discriminator and suffer from training instability (mode collapse), which is fatal for generating precise weight parameters where stability is paramount.
    - **VAEs**: Similar to MLPs, VAEs tend to produce "blurry" results by averaging the distribution. In weight space, an "averaged" weight vector often fails to function correctly.
3. **Why Diffusion**: Diffusion models excel at fine-grained, high-fidelity generation. As supported by recent works like p-diff[1], diffusion is superior in modeling the complex distribution of neural network parameters, ensuring the generated weights lie strictly on the valid solution manifold.

**[Common Concern 4] Comparison with SOTA (AlphaEdit) (Raised by R2)**

- **Response**: We compared MeG with the recent AlphaEdit (ICLR 2025) on Phi-2-ZsRE from 1024 to 10k editing scale.
- **Result**: Our **MeG significantly outperforms AlphaEdit**, particularly in Locality at all editing scales ( e.g., 91.14% vs. 9.87% under AG generation setting or 95.84% vs. 56.32% under TF generation setting at 10k scale).
- **Analysis**: The performance gap stems from how each method handles unrelated knowledge:
    - **AlphaEdit's Limitation**: a. It constructs constraints using a fixed set of ~100k random Wikipedia entries to represent "unrelated knowledge." However, the true scope of unrelated knowledge (Total LLM Knowledge minus Edited Knowledge) is vastly larger. This subset is insufficient to constrain the optimization, leading to significant bleed-over into general knowledge. b. AlphaEdit relies on a fixed "null space" from unrelated data, which saturates at large scales, leading to a significant Reliability drop (only 72.43% in 10k editing scale).
    - **MeG's Advantage**: a. MeG employs a **Familiarity Network** that assigns random low-entropy fingerprints to edited knowledge. We utilize an **entropy criterion** to distinguish unrelated queries: unrelated inputs naturally yield high entropy distributions, while edited inputs yield low entropy. This is a robust routing mechanism that requires **only the edited knowledge** to function, eliminating the need to sample from the massive "unrelated" space. b. MeG’s dynamic generation and routing mechanism remains robust regardless of scale. (maintain 95.07% Reliability and 91.14% Locality in 10k editing scale).

Detailed responses to specific questions (e.g., entropy threshold sensitivity, training costs) are provided in the individual threads. We believe these results demonstrate that MeG is a robust, scalable, and practical solution for massive knowledge editing.

[1] Wang K, et al. Neural Network Diffusion[J]. CoRR, 2024.

---

### Author Response · Authors · 2025-12-03
**General Response (1/2)**

Dear AC and Reviewers,

We sincerely thank the AC and all reviewers (R1-4YiG, R2-tF5C, R3-Upk2, R4-2bpq) for their time and constructive feedback. We are encouraged that reviewers unanimously recognized the **novelty** of reframing massive knowledge editing (KE) as conditional weight generation via diffusion and acknowledged our **strong empirical performance**, particularly in **Locality** and **Scalability** up to 10k edits.

We have uploaded a revised paper with changes highlighted in **blue**. Below, we summarize the major updates and point to their specific locations in the revision:

1. **New Baseline Comparison**: Added comparison with AlphaEdit [ICLR'25] (**Appendix A.7.1**).
2. **Ablation Studies**: Added comparison between DiT and MLP as weight generation module (**Section 6.2**).
3. **Capacity Analysis**: Added experiments on UnKEBench (Long-text) and MQuAKE (Multi-hop) (**Appendix A.6**).
4. **Further Efficiency Analysis**: Added detailed training cost and inference throughput analysis (**Appendix A.8**).
5. **Robustness Checks**: Added sensitivity analysis of entropy threshold (**Appendix A.5**).

Below, we address the common concerns raised by multiple reviewers.

**[Common Concern 1] Inference Efficiency and Complexity (Raised by R1, R3, R4)**
Reviewers expressed concern about the complexity of the pipeline and the latency of the diffusion process.

- **Response**: While our method involves a generative step, it is **more efficient in practice** for massive editing tasks than baselines.
1. **Higher Throughput (Queries Per Second, QPS)**: As shown in our new experiments (Table below), MeG achieves **2x-3x higher Queries Per Second (QPS)** than MEMIT and MALMEN at scale. This is because baselines suffer from "catastrophic collapse" at massive-editing scenarios, generating excessively long, repetitive garbage sequences, whereas MeG maintains concise and correct generation.
2. **Fast Sampling**: We utilize **DDIM** to compress sampling to 50 steps, adding only ~5% overhead compared to a standard forward pass.
3. **Selective Routing**: The Familiarity Network filters out irrelevant queries, incurring zero diffusion overhead for general knowledge.

**Table: Inference throughput (QPS) of different KE methods on various LLMs.**
|LLM(Phi-2)|MEMIT|PMET|MALMEN|MeG (Ours)|
|-|-|-|-|-|
|**Phi-2**|2.38|2.48|1.38|**6.82**|
|**GPT-J**|1.32|1.29|0.85|**2.75**|
|**Llama-3**|1.55|1.54|1.05|**3.70**|

**[Common Concern 2] Is a Single Neuron Sufficient? (Raised by R2, R4)**
Reviewers questioned if adding a single neuron limits the capacity for complex tasks like multi-hop reasoning or long-text generation.

- **Response**: Our new experiments confirm that a single neuron is sufficient to act as a "steering vector" for complex tasks.
1. **Multi-hop Reasoning (MQuAKE)**: We have tested MeG on MQuAKE-CF. MeG achieves a multi-hop accuracy of 41.41%, significantly outperforming MEMIT (0.10%) and MALMEN (5.66%), proving it can generalize to complex reasoning.
2. **Long-text Generation (UnKEBench)**: On tasks requiring ~100 token generation, fine-tuning a single neuron achieves 98.6%+ Reliability across Phi-2, GPT-J, and Llama-3.
3. **Theoretical Insight**: Recent "Linear Representation Hypothesis"[1,2] studies suggest knowledge is often encoded linearly; thus, a single neuron is sufficient to shift the activation trajectory.

[1] Park K, et al. The Linear Representation Hypothesis and the Geometry of Large Language Models[C], ICML,2024.

[2] Zou A, et al. Representation Engineering: A Top-Down Approach to AI Transparency[J], CoRR, 2023.

---

### Meta-Review · Area_Chair_Kns1 · 2026-01-07

**Summary:**

Four types of concerns can be summarized below:
1. Methodology Complexity and Justification for Diffusion (Reviewer 4YiG and 2bpq). Multiple reviewers questioned the necessity of using a Diffusion Transformer (DiT). They asked for justification on why a simpler mapping function (like an MLP) or other generative models (like GANs or VAEs) could not be used instead of the potentially unstable and expensive diffusion process.
2. Inference Latency and Training Costs (Reviewer Upk2 and 2bpq). They pointed out that running a 50-step diffusion model for every relevant query introduces substantial latency compared to baselines like MEMIT, which only require a standard forward pass. In addition to inference latency, they also raised questions about the cost of offline training and the effectiveness of sampling.
3. Theoretical Capacity and Robustness (Reviewer tF5C and 2bpq). They questioned whether a single neuron is sufficiently expressive to handle complex knowledge updates, such as multi-hop reasoning or long-form generation, suggesting it might be inadequate compared to the storage capacity required.
4. Familiarity Network (Router) Reliability (Reviewer Upk2 and 2bpq). They asked if it could reliably distinguish unseen correct facts (general knowledge) without misclassifying them as "irrelevant" or vice versa, which is critical for maintaining locality.

**Reviewer Concerns:**

The reviewer's concerns may be addressed by the rebuttal:

- The authors successfully provided additional data regarding inference latency (responding to R1, R3) by demonstrating that using DDIM with 50 steps yields acceptable QPS compared to MEMIT. They also added an ablation study comparing the DiT to an MLP (responding to R1, R4), and included a comparison with AlphaEdit (responding to R2), which was a necessary addition.

The reviewer's concerns may still be outstanding:

- Methodological Complexity and Practicality  (Reviewer 4YiG and 2bpq): The proposed MeG pipeline is excessively complex and computationally heavy compared to the problem it solves. It requires three distinct training/fine-tuning phases for a specific batch of edits: (1) InfoNCE tuning of the encoder, (2) Training the Familiarity Network, and (3) The "Knowledge-Weight Collection" followed by training the DiT. As noted by the reviewers, this "offline" cost is massive. The method effectively shifts the burden from "editing" to "training a generator on the specific edit set," which undermines the utility of knowledge editing as a lightweight alternative to fine-tuning.

- Lack of Generalization / The "Lookup Table" Issue (Crucial Flaw): This is a critical soundness issue that was not fully resolved. The method operates by learning a mapping from specific queries to specific pre-optimized weights. This functions as a sophisticated, over-parameterized lookup table rather than a generalized editing mechanism.

- Baselines and Model Selection: While Llama-3 was used, a significant portion of the evaluation relies on outdated architectures (Phi-2, GPT-J). Secondly, the proposed method bears a striking resemblance to T-Patcher. While the authors discuss the distinctions textually, the absence of an empirical comparison with T-Patcher is a significant omission.

**Reviewer Scores:**

Overall, I believe the four reviewers will at least maintain their current scores.

---

### Decision · Program_Chairs · 2026-01-26

Accept (Poster)